# The Neverwhere Visual Parkour Benchmark Suite

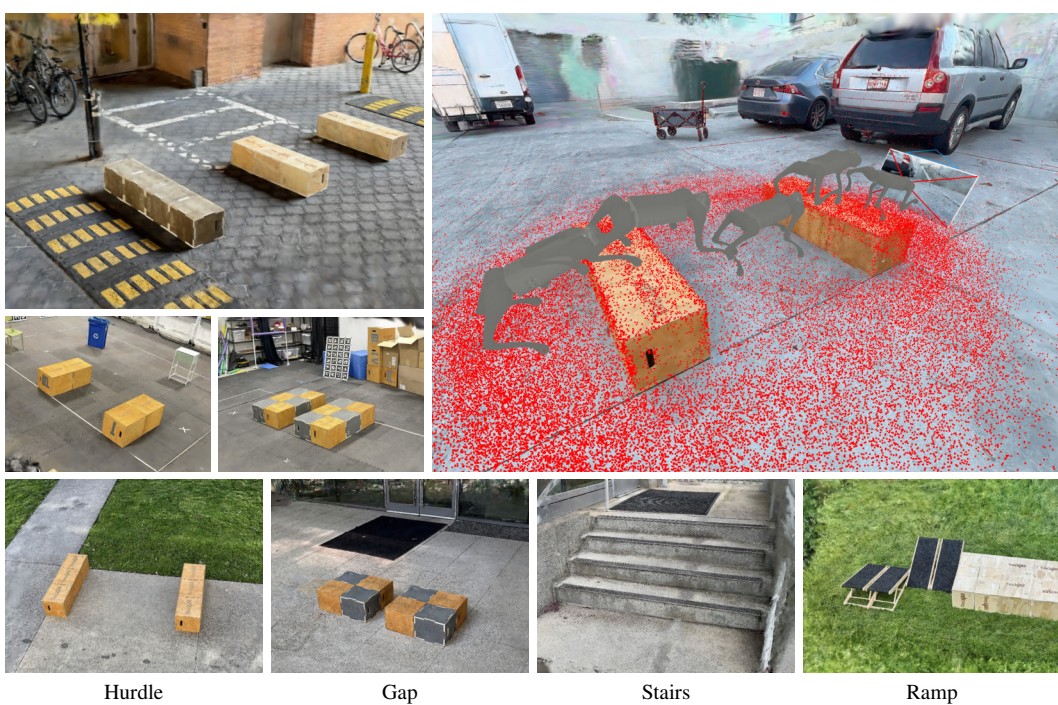

| Hurdle | Gap | Stairs | Ramp |

Figure 1: **The Neverwhere Benchmark Suite.** We offer over *sixty* high-quality Gaussian splatting-based evaluation environments, and the Neverwhere graphics tool-chain for producing accurate collision mesh. Our aim promote reproducible robotics research via fully automated, continuous testing in closed-loop evaluation.

## Abstract

State-of-the-art visual locomotion controllers are increasingly capable at handling complex visual environments, making evaluating their real-world performance before deployment increasingly difficult. This work intends to narrow this train/evaluation gap by developing a collection of hyper-photo-realistic, closed-loop evaluation environments – The Neverwhere Benchmark Suite – comprised of over sixty 3D Gaussian Splatting of urban indoor and outdoor scenes. Our goal is to encourage large-scale and reproducible robot evaluation by making it easier to create and integrate Gaussian splats-based reconstructions into simulated continuous testing setups. We also underscore the potential pitfalls of relying exclusively on 3D Gaussian-generated data for training, by providing policy checkpoints trained over multiple Neverwhere scenes and their performance when evaluated in novel scenes. Our analysis illustrates the necessity of sourcing diverse data to ensure performance. Anonymous Website: link

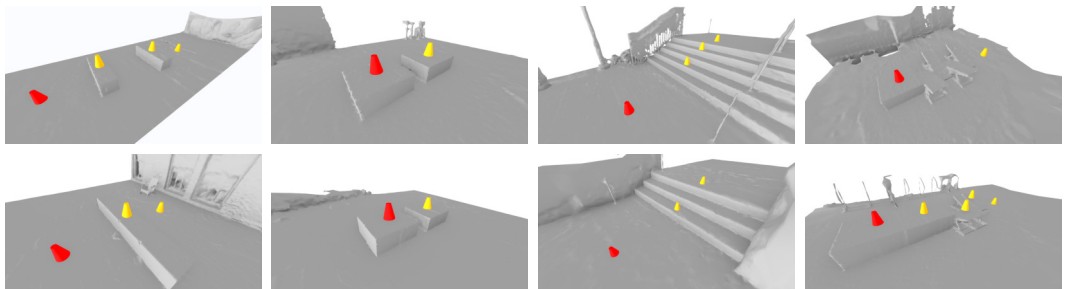

Figure 2: **Task Waypoints Layout.** From left to right: Hurdle, Gaps, Stairs, Ramp. The robot starts from the right. **Yellow cones** indicate waypoints, whereas **red cones** are the final goal.

# 1  Introduction

The past few years witnessed a rapid acceleration in progress in robotics. Data-driven, general-purpose learning algorithms, which treat specific tasks as data points from a general problem class, proved to be the scalable approach to producing robots that are robust, capable, and intelligent. As our robots graduate the confined lab environments to face the open world, real-world evaluation and hand-crafted simulation environments are proving insufficient. We need an evaluation strategy that is equally scalable to quantify progress. How do we create abundantly diverse and realistic environments to test our robot?

This work aims to develop a scalable approach to testing real-world visuomotor policies in automated, closed-loop simulations. We focus on visual locomotion in legged robots as our test bed, a class of robotic tasks where perception is tightly coupled with actions. Our intention is to start with a domain where the 3D environment is complex, but the physics is relatively simple. Our main contribution is Neverwhere (see Fig. 1), a collection of over 60 high-fidelity digitally recreated scenes that covers diverse urban structure, including stairs, speed bumps, indoor carpeted lab spaces and the outdoor, with and without vegetation. An equally essential objective is to empower the community to build their own set of benchmarks.

The Neverwhere tool-chain address three essential challenges in building evaluation environments for robots: The primary challenge is to capture the world in its full messiness which exceeds the expressivity of traditional 3D mesh. The second challenge is that in practice, quadruped robots observe the world from an angle that sits out of the distribution of human camera views. This coupled with the extraordinary expressivity of the Gaussian substrate, results in poorly rendered ego camera input. The final challenge is about geometry. It remains difficult, in practice, to obtain detailed collision mesh from 3D Gaussian that are modeled using hand-held iPhone videos. We solve all three challenges, by developing a better initialization scheme that takes advantage of traditional multi-vew stereo reconstruction (Sec. 4).

Our contributions are summarized as follows:

- We introduce the Neverwhere benchmark suite, featuring over 60 high-quality environments powered by 3D Gaussians, encompassing a diverse range of urban indoor and outdoor scenes.
- We present a data collection toolchain that facilitates the generation of new benchmark environments with minimal human intervention, allowing users to create reconstructed scenes directly from uncalibrated images or videos.
- We provide and release visual parkour policy checkpoints trained directly on the Neverwhere 3D Gaussian environments, offering baseline results to support further research and exploration.

# 2  Related Works

Robotics research has a long tradition of using physics simulation engines to evaluate planners and policies prior to real-world deployment [6, 27]. More recently, improvements in learning-based approach has enabled neural controllers to directly map high-dimension visual data into joint configurations [7, 5] while also expanding along the dimension of the *number* of skills learned by a

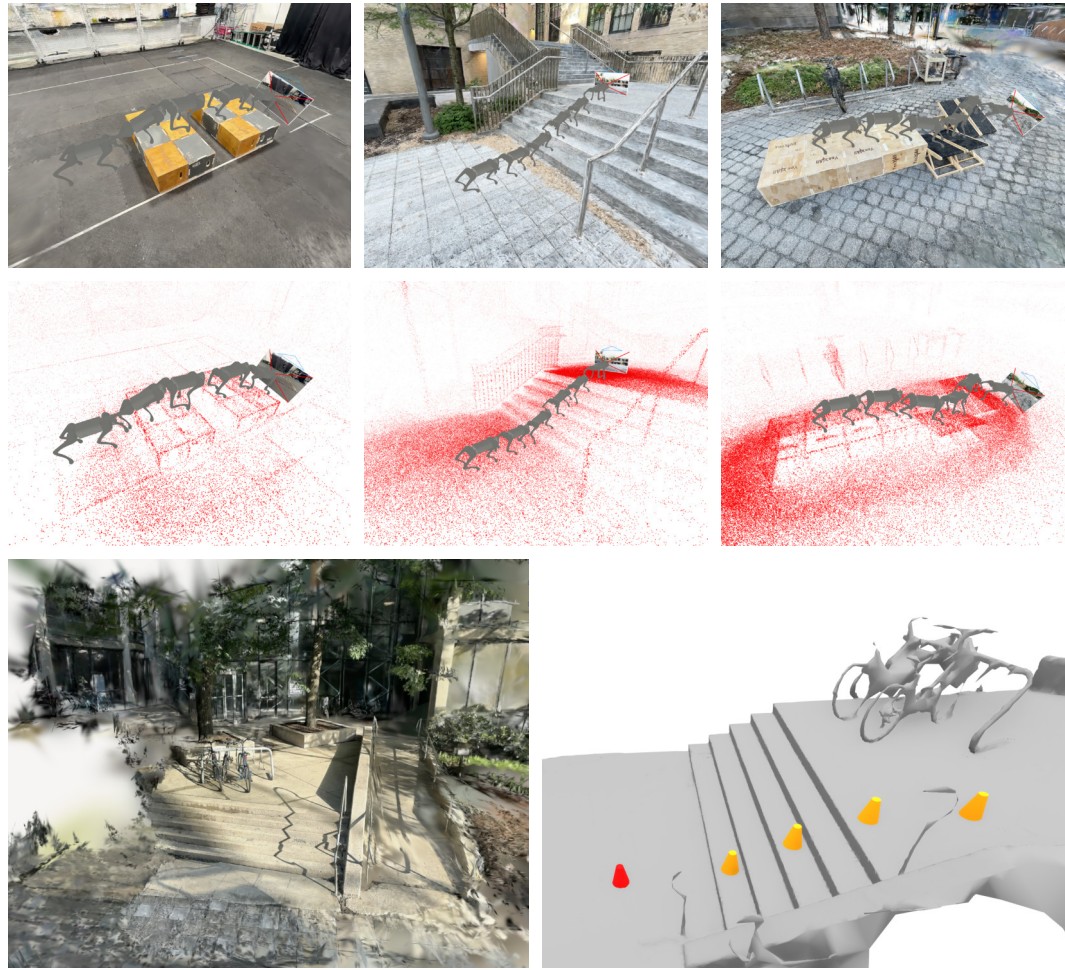

Figure 3: **Example Scenes and Collision Geometry.** Showing a subset of the environments, on four parkour locomotion tasks: Hurdles, Gaps, Stairs, and Ramps. The bottom row displays annotated trails with labeled waypoints that define each evaluation task. All images are rendered.

single, general-purpose control policy [34, 1]. In contract, the rendering setup used by early robotics benchmarks [11] was relatively primitive as the intended purpose was for humans to visualize and debug failure, as opposed to simulating accurate camera sensor readings for a policy.

Both the quality, and the range of physics and material have been improved significantly in more recent physics engines such as IssacSim [22] and ManiSkill3 [8, 26], extending coverage to deformable material, fluid, and caustics. Despite these improves, lack of 3D content remains a bottleneck. Recent benchmark efforts significantly raised the bar: BiGym [4], for instance, provided a high-quality, manually CAD'ed 3D collision mesh for an articulated dish washer. At the scene level, RoboCasa [20] provides fourteen manually designed kitchen scenes.

Neverwhere differs from these prior efforts [21, 16, 10, 15, 4] in two ways: First, advancements in neural scene representation made investment in traditional assets and lighting setup less critical. Neverwhere uses 3D Gaussian Splatting to replace mesh-based rendering, which not only simplifies construction but also enables the creation of highly detailed digital replicas, which can be done by the end-user using Neverwhere's open-source toolkit. Second, Neverwhere builds upon the MuJoCo [27] physics engine and aims to provide accurate collision geometry. It does so without requiring LiDAR sensors and depth measurements. The closest are simulators from autonomous driving that are used for closed-loop evaluation of self-driving vehicles (SDV). Among them, UniSim [30] uses detailed mapping data to create the environment and replay pre-recorded driving episodes to produce safety-critical scenarios involving pedestrians and other vehicles. Neverwhere is similar in spirit but currently lacks UniSim's advanced scene decomposition and the ability to animate other actors.

Nevertheless, our goal is to use Neverwhere as a foundation for building open-source, customized evaluation setups for robotics, with plans to integrate more sophisticated capabilities in the future.

Generative AI is increasingly being used in robotics to generate task scenarios, rewards, and assets. Platforms such as RoboCasa, RoboMimic, and MimicGen [20, 29, 18] employ generative models to diversify training environments. However, Neverwhere takes a different approach by emphasizing photo-realism and accurate modeling over diversity. We argue that when it comes to evaluation benchmarks, a deterministic, curated set of challenging test cases prioritizes signal over noise. Neverwhere's use of 3D Gaussian splats is better suited for evaluation than for training, as the former favors reduced variability.

# 3 Tasks and Reinforcement Learning Setup

Neverwhere offers four types of parkour tasks. For each, there is a group of physical environments, created from 3D scans of two university campuses, featuring both outdoor and indoor domains with different obstacle layouts and appearances. The tasks were written in Python, and physics simulation is implemented with MuJoCo [27]. The locomotion setup follows:

**Action.** The action space consists of twelve target joint positions for the quadruped robot, with each of the robot's four legs having three actuated joints: hip abduction/adduction, hip flexion/extension, and knee flexion/extension.

**Observation.** The observation includes the robot's ego state, $\mathbf{e} = \{v, \mathbf{q}, \dot{\mathbf{q}}\}$, where $v$ is the linear velocity, $\mathbf{q}$ the joint positions, and $\dot{\mathbf{q}}$ the joint velocities, along with the robot's previous actions. For evaluating visual policies, we provide visual observations in different data modalities, including RGB renders from gsplat, depth maps, point clouds, and semantic maps. The rendering pipeline detailed in Sec. 3.2 provide these diverse data modalities for visual policies' input.

**Privileged Observation.** This includes a heightmap of the scene, offering a top-down view of the terrain, which can further be processed into ScanDots observation for lightweight inference [3]. The moving direction, represented as a single angle value, can also be provided to guide navigation.

## 3.1 Parkour Tasks

We designed four challenging scenarios to evaluate a robot's ability to generalize locomotion skills, adapt to varying terrain, and handle physically demanding tasks, following the task design in [3](introducing in **ascending order of difficulty**.): 1) overcoming obstacles of a certain height (**hurdles**), 2) jumping across gaps of varying lengths (**gaps**), 3) navigating sloped surfaces (**ramps**), and 4) walking up stairs (**stairs**).

We labeled waypoints to define a specific trail in each scene, outlining the exact task and target for the robot. We then measured the robot's performance by calculating two metrics: the **Success Rate**, determined by the percentage of waypoints reached, and **X displacement**, the total distance moved in the +X direction. The following section provides detailed task definitions and the common waypoint distribution for each task. Illustrations of the waypoints are listed in Fig. 3.

**Hurdles.** Most hurdle scenes are manually constructed. Each hurdle consists of several boxes arranged side by side to form a low barrier. Typically, 1 to 3 such barriers are placed consecutively within a scene, with waypoints labeled on top of each one. In addition to these manually created scenes, a small portion of the dataset uses natural outdoor elements, e.g. long stone benches, as hurdle obstacles.

**Gaps.** All gap scenes are manually set up, either indoors or outdoors. We construct two box-based platforms with a 12-inch or 16-inch gap between them to simulate jumping over a gap. Waypoints are labeled on top center of the platforms.

**Ramps.** Ramp scenes are designed to test the robot's ability to walk on inclined surfaces. Around four sloped boards are placed in a staggered arrangement alongside a raised platform built from boxes. Waypoints are primarily placed at the end of each trail.

**Stairs.** Stair scenes involve real-world staircases of various heights, materials, and textures, captured both indoors and outdoors. The robot is tasked with climbing up the stairs, and waypoints are placed along the steps.

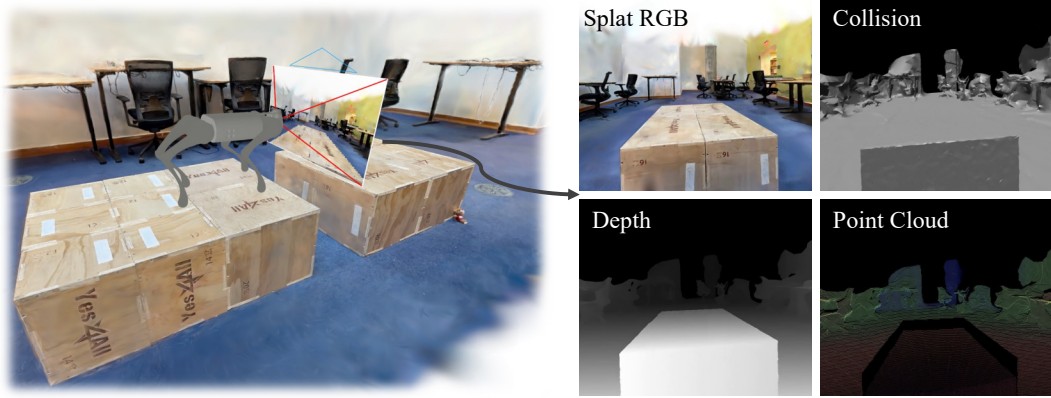

Figure 4: **Rendering Wrappers.** We designed a rendering pipeline that provides diverse wrappers for multi-modal observations, including but not limited to: color images from Gaussian splats, depth maps, heightmaps, and LiDAR projections, to support a wide range of visual based policies.

## 3.2 Rendering Wrappers

**Gaussian Rendering.** We adopt 3DGS [12] for photo-realistic rendering to reduce the domain gap in policy training and evaluation. 3DGS provides high-quality visual observations and real-time performance, making it well-suited for close-looped evaluation. To incorporate visual targets (e.g., cones) that are not present in the original 3DGS scene, we blend them into the rendered view using semantic masks rendered from MuJoCo [27], enabling consistent integration of task-relevant cues. Through experiments (in Sec. 5.3), we found visual cones are strong visual cues that lead to better policies.

**Depth.** Depth is obtained by converting MuJoCo-rendered maps [27] into MiDaS-style inverted depth [23]. These depth maps serve as observation inputs for robot policy learning. They can also be used as conditioning inputs for depth-conditioned generative models [33], enabling robust training and zero-shot transfer to real-world RGB inputs [31].

**Semantics Mask.** We generate semantic masks by grouping objects based on naming rules, enabling simple target-background segmentation for downstream tasks.

**Heightmap.** A top-down bird's-eye-view heightmap is rendered to capture terrain geometry while excluding movable objects, aiding navigation and privileged policy training.

**LiDAR Projection.** Simulated LiDAR rays produce point clouds based on scene geometry, supporting policies that rely on spatial awareness and obstacle detection.

## 4 From Photons to Splats: The Neverwhere Environment Builder

We aim to develop a scalable and efficient toolchain for creating benchmark scenes, enabling users to easily generate their own digital twins. Existing 3DGS techniques [13, 35, 14, 28, 32, 19, 9, 17] excel at visual fidelity, but the resulting meshes may lack the physical accuracy required for reliable physics-based interactions. Our robot needs precise collision geometry to be evaluated correctly. Developing high-fidelity digital replicas for robot simulation necessitates both high visual quality with minimal domain gap and accurate physical modeling of real-world geometry to facilitate robot interaction. By leveraging the power of robust and efficient SfM and MVS modules within a unified pipeline, Neverwhere automatically converts pose-free multi-view images into registered pairs of 3D Gaussians and high-quality collision geometry. Thus, by combining the strengths of 3D Gaussians for appearance representation with spatially aligned meshes for the robot simulation platform, we construct a complete physics-aware robot simulation environment.

In detail, given a set of $N$ uncalibrated images $\mathcal{I} = \{\mathbf{I}_i\}_{i=1}^{N}$, a camera pose estimation module $\Theta$ is used to estimate their poses (Fig. 5-(2)), yielding $\mathcal{P} = \{\mathbf{P}_i\}_{i=1}^{N}$. Subsequently, a mesh reconstruction

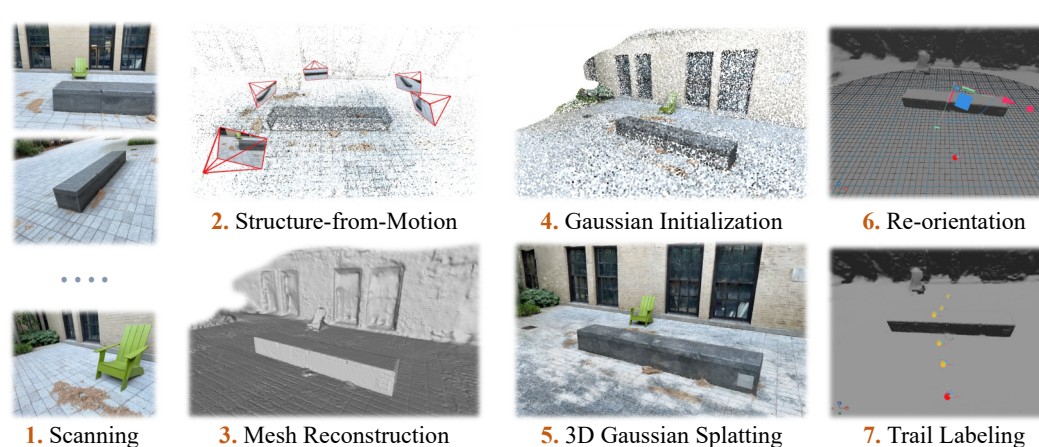

**2.** Structure-from-Motion     **4.** Gaussian Initialization     **6.** Re-orientation

**1.** Scanning     **3.** Mesh Reconstruction     **5.** 3D Gaussian Splatting     **7.** Trail Labeling

Figure 5: **Neverwhere Toolchain.** The toolchain takes (1) multi-view images as input and follows a sequential process: (2) A Structure-from-Motion modules is applied to obtain camera calibrations, (3) An optimization-based MVS module is used to estimate the scene geometry, (4) Points are sampled from the textured meshes, which are then used as initializations for (5) training 3D Gaussians to model the scene. Once the scene modeling is complete, human input is required for (6) reorienting the mesh to align with scene conventions and (7) labeling waypoints for visual parkour policies.

module $\Phi(\mathcal{I}, \mathcal{P})$ is employed to recover the scene mesh (Fig. 5-(3)). In our reconstruction toolchain, COLMAP [24, 25] is utilized as $\Theta$ for camera calibration, and OpenMVS [2] serves as $\Phi$ to process the calibrated poses and images into a fine-grained collision mesh $\mathcal{M}$.

Following this geometric reconstruction, we optimize 3D Gaussian Splats for scene appearance modeling (Fig. 5-(5)). Vanilla Gaussians are prone to overfitting to training views, often resulting in suboptimal novel view rendering quality when the viewpoint significantly deviates from the training data (e.g., views of a quadruped robot versus typical handheld camera views). To address this, we introduce geometrical constraints to achieve better novel view rendering. Therefore, we extract depth maps $\mathcal{D} = \{\mathbf{D}_i\}_{i=1}^{N}$ and confidence maps $\mathcal{C} = \{\mathbf{C}_i\}_{i=1}^{N}$ for each image in $\mathcal{I}$ from the patch-matched geometric cache of $\mathcal{M}$. These maps are then used to supervise 3D Gaussian training with the following loss term:

$$\mathcal{L} = (1 - \lambda_r) \left\| \mathbf{I} - \hat{\mathbf{I}} \right\|_1 + \lambda_r \mathcal{L}_{\text{SSIM}} + \lambda_D \left\| \mathbf{C} \odot (\mathbf{D} - \hat{\mathbf{D}}) \right\|_1 \tag{1}$$

Here, $\lambda_D$ is the weight for depth supervision. In practice, we initialize Gaussians with sampled colored points (Fig. 5-(4)) from the collision mesh $\mathcal{M}$, as this provides improved geometric priors compared to the standard initialization using SfM points.

SfM methods recover scene geometry up to an arbitrary scale and orientation for uncalibrated images. Thus, the reconstructed Splats and Mesh are not aligned with real-world scale or the z-up convention, making them unready for robot simulation. Additionally, evaluating parkour policies requires task definitions within the scene, such as waypoint trails to measure robot success rates. To address this, we developed an intuitive labeling tool that lets users quickly reorient and rescale the mesh to match real-world coordinates (Fig. 5-(6)) and define tasks by labeling waypoints (Fig. 5-(7)). After labeling, the system automatically generates simulation task configurations for the benchmark environments. The full process takes about two minutes (see supplementary materials for demo).

**Improving 3DGS Modeling from Hand-held iPhone Videos.** Accurate collision geometry is essential for precise contact and physics simulation. The typical solution involves using multi-view stereo methods to generate this geometry. However, geometry generated by mobile applications often lacks the details required for fine-grained contact simulation due to the limited computational power of mobile devices. Furthermore, these methods often require a depth sensor for robust reconstruction. We propose an alternative approach using OpenMVS to reconstruct physical geometry with enhanced quality and provide additional cache to improve 3DGS quality. This method produces high-quality, cost-effective geometry without requiring external sensors. As shown in Fig. 6, the meshes generated

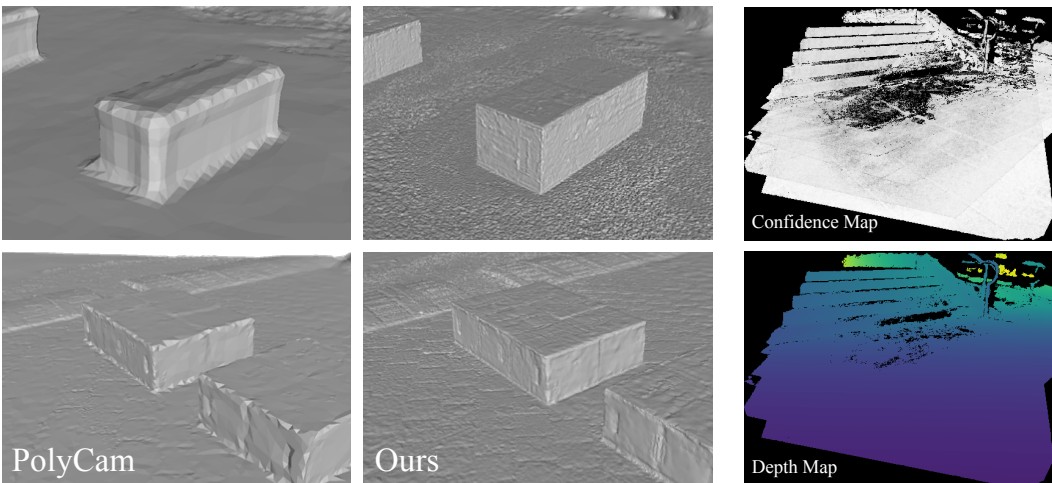

Figure 6: **OpenMVS provides detailed geometry.** (left) mesh taken from a consumer 3D scanning app PolyCam. (right) reconstruction from OpenMVS.

Figure 7: **Confidence and depth maps.** We extract these from the SfM pipeline.

by OpenMVS [2] exhibit fine geometric details and provide good scene coverage, matching or even surpassing the results from on-device mobile software.

**Geometry Guided Gaussian Initialization.** To achieve higher-quality 3DGS with improved geometry and consistent depth, we use colored points sampled from the textured mesh produced by OpenMVS [2] for initialization (Fig. 5-(4)). This approach is more geometrically organized than using scattered points from SfM alone, as illustrated in Fig. 8, resulting in better 3DGS for rendering geometry-consistent views from robot's views.

# 5 Experiments

Our design intention is for Neverwhere to be used as part of an automated, continuous testing setup that quickly and scalably uses closed-loop simulation to assess the policy before its real-world deployment. Training and testing environments have different requirements: the former benefits from system coverage and entropy, whereas the latter is better conducted deterministically to maximize interpretability. To explore the capabilities of the Neverwhere benchmark under constrained entropy and limited scenes, we conducted experiments on closed-loop training to provide additional insight. Although the results indicate limited generalization that restricts effective closed-loop training in this context, the benchmark consistently reflects the robot's capabilities from an evaluation standpoint.

## 5.1 Training Setup:

We performed closed-loop training using a teacher-student behavior cloning approach. We re-trained a privileged teacher policy from [3] to provide guidance. We trained both depth-based and RGB-

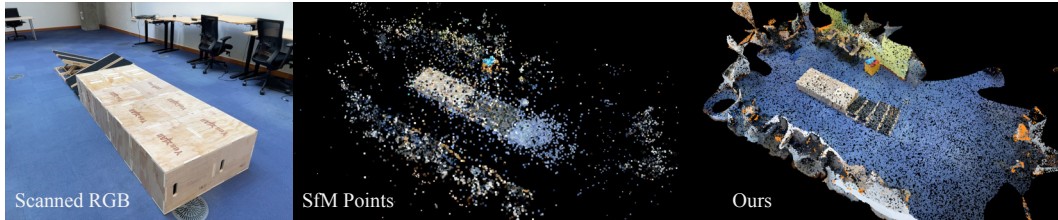

Figure 8: **Gaussians Initialization**. Instead of using SfM points, we utilize colored points sampled from the reconstructed textured mesh generated by our pipeline for initialization, ensuring improved geometric accuracy of the Gaussians.

based visual policies (rendered via a Gaussian Splats wrapper) through on-policy learning. Dataset aggregation (DAgger) was employed, sampling 1,000 trajectories per DAgger iteration. The teacher policy architecture follows the approach described in [3]. For the student policies, we adopted Action Chunking Transformers (ACT) to improve their ability to handle challenging tasks such as gaps and ramps. We tried two settings:

**(1) Single-scene Training:** We trained the policy on a single environment and performed 4 DAgger iterations. We evaluated its performance both on the training scene itself and on other scenes within the same task. Domain randomization was turned off for this setup.

**(2) Multi-scene Training:** We split 70% of the scenes from a specific task to create a training set, and the remaining 30% were used for evaluation. The goal was to explore the policy's ability to transfer across different simulated environments within the same task. For these experiments, we applied domain randomization during training: **I. Depth visual policy**, we added random noise to the depth maps and randomly zeroed some pixels. **II. RGB visual policy**, we applied random rotations, cropping, Gaussian blur, and color transformations to the input images.

All experiments were conducted using the Unitree Go1 robot, with physics simulation powered by MuJoCo.

## 5.2 Single-Scene Closed-Loop Training

We first test whether a visual policy can fit well in a single domain, to verify both the robot's learning ability and the effectiveness of our digital environment. We select three tasks in increasing order of difficulty: Hurdles (easy), Gaps (medium), and Stairs (hard). The policy is trained on one scene from each task and evaluated on all other scenes within the same task. To ensure variation, random noise is added to the trajectory, making the evaluation trails different from the training ones. As shown in Fig. 9, the policy performs well on the training scene but generalizes poorly on unseen scenes, which matches our expectation as the training domain is limited. One interesting finding is that performance slightly improves on some unseen scenes that share similar visual characteristics (e.g., both being outdoor environments) with the training domain, as observed in the bottom-right corner of each confusion matrix in Fig. 9.

## 5.3 Multi-Scene Closed-Loop Training

Given that policies fit well in a single domain, we further investigate whether our scenes support effective closed-loop training for visual policies. We evaluated on both training set and evaluation set, the performance gap between training and evaluation sets is large (about 50% on average) for the Stairs task Fig. 10-(A), but relatively small (about 10% on average) for the Gaps task Fig. 10-(B). This suggests that the trained visual policies exhibit limited generalization on our benchmark, particularly for more challenging tasks.

**Ablation on Observation Types:** The above experiments do not use include those cones as observation. We further investigate how different observation types affect policy performance. (1) RGB vs.

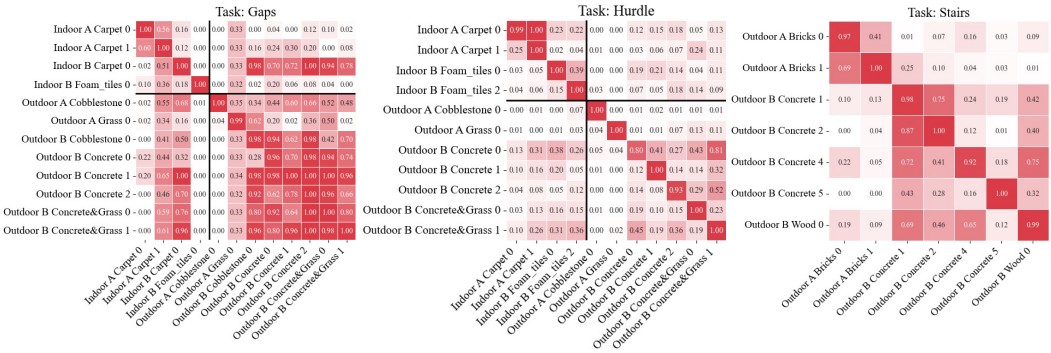

Figure 9: **Generalization of Single-Scene Policies.** Each policy (row) is trained in a single environment. The cross-scene generalization shows clear clustering.

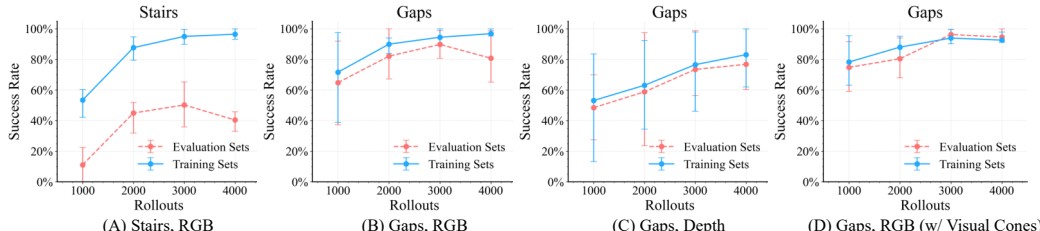

Figure 10: **Results on Multi-Scene Closed-Loop Training.** We split each task's scenes into 70% for training and 30% for evaluation. For each scene, we perform 50 rollouts and report the average success rate over all rollouts in the train and evaluation sets. See the supplementary material for visual references of the listed scene names.

Depth: As shown in Fig. 10-(B) and (C), both inputs are trained with domain randomization. While RGB yields moderate results, Depth performs poorly even on the training set, suggesting that depth represented in our benchmark is currently less effective for learning. (2) With vs. without visual cones: Comparing Fig. 10-(C) and (D), adding visual cones effectively improves training efficiency and overall performance on both training and evaluation sets. This highlights the benefit of consistent, explicit visual cues (e.g., cones) in aiding policy learning under diverse visual domains.

## 5.4  Evaluating Visual Parkour Policies

Neverwhere is designed to test robot policies before real-world deployment. We evaluate visual policy checkpoints trained by *Lucidsim* [31], analyzing their performance gap between simulation and real environments. Results show that Lucidsim achieves reasonable success rates, with some scenes exceeding 95% and most scenes above 50%. This aligns roughly with Lucidsim's reported results of 73.3% for hurdles and 100% for stairs. Note that the real-robot test environments differ from our benchmark, so performance differences are expected.

Table 1: Evaluating *Lucidsim* [31] with Our benchmark.

| Tasks | Scenes | Rollouts | Average | Highest | Median | Lowest |
|---|---|---|---|---|---|---|
| hurdle | 15 | 50 | 59.67% | 95.33% | 68.67% | 0.00% |
| stairs | 14 | 50 | 55.82% | 93.16% | 55.37% | 2.78% |

## 6  Conclusion

We proposed the Neverwhere benchmark suite along with a real-to-sim toolchain. Our goal is to provide the community with a practical tool for testing policies before real-world deployment, potentially as part of a continuous testing setup. This work aims to accelerate the development of scalable and efficient approaches for robot evaluation, as current robot policies are becoming increasingly capable while existing evaluation methods remain inefficient.

Although the Neverwhere toolchain was initially designed for our locomotion evaluation benchmark suite, its capability for creating contact-aware real-world digital twins is broadly applicable across various domains. This unified framework, built on freely accessible pipelines, is designed to support the real-to-sim-to-real research community.

**Limitations.**  Although Neverwhere has provided over 60 diverse scenes, expanding the benchmark with additional diverse scenes requires further human intervention and effort, as the reconstructed 3D splats and meshes are not automatically aligned with real-world scale or the standard z-up orientation. This necessitates manual reorientation, rescaling, and task labeling before they can be used in robot simulations. Our future work will explore learning-based methods for automatic alignment and scene labeling of the 3D reconstructions.

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
