# Supplementary Materials

# The Neverwhere Visual Parkour Benchmark Suite

## Contents

## 1 Additional Experiments

This section extends the experiments in the main draft. The first two subsections present additional results from multi-scene closed-loop training and analyze the policy's performance and its ability to generalize to unseen scenes after training on our benchmark. The final two subsections provide ablations on different observation types, examining how various observation cues impact policy performance, including: (1) **Full Observation**: includes rendered visual cones and waypoint directions in the observation space; (2) **Without Visual Cones**; (3) **Without Direction Information**.

### 1.1 Task-Specific Closed-Loop Training

Figure 1 shows the results of closed-loop training for each individual task. Each policy is trained on around 10 scenes of the same task with 4 DAgger rounds. We report performance on both the training and evaluation sets (See Tab. 2 for the scene list), the observation type is full Observation. The results reveal a significant performance gap between the training and evaluation sets. This disparity indicates that closed-loop training is not effective on our benchmark, likely due to the limited number and diversity of scenes required for robust policy generalization. Nonetheless, as an evaluation benchmark, Neverwhere plays a valuable role in assessing robot policies before real-world deployment.

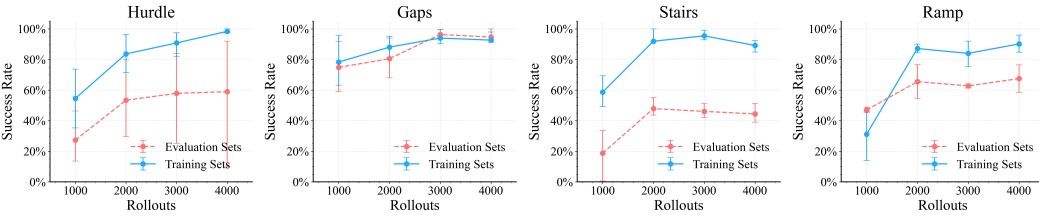

Figure 1: **Task-Specific Multi-Scene Closed-Loop Training. Observation Type: RGB, Full.**

### 1.2 Combined Task Closed-Loop Training

We further investigate whether combining scenes across all tasks, thus increasing both scene and task diversity, can yield a more generalizable policy. We create a unified training set by merging all scenes from each task. Results are shown in Figure 2. The policy achieves about 85% success on the training set, but exhibits a 30% performance drop on the evaluation set. This large gap suggests that, despite increased diversity, the scene set remains insufficient for training a robust visual policy. Real-world robot policy training typically requires significantly larger and more varied datasets.

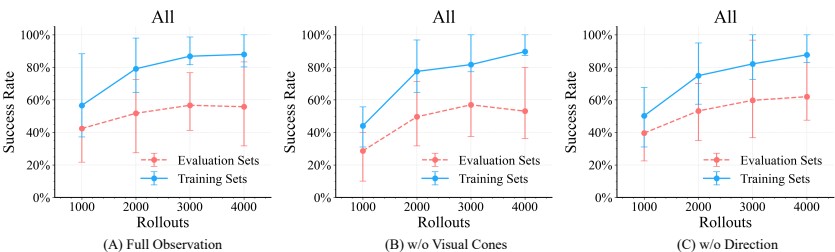

Figure 2: **Combined Task Multi-Scene Closed-Loop Training. Observation Type: RGB.**

## 1.3 Ablation on Different Observation Cues

**Effect of Visual Cones.** Visual cones blended into RGB images may help the robot follow targets, but they represent artificial cues not typically available in real-world settings. To better align with real deployment conditions, we ablate this feature. Comparing performance on two tasks: Hurdle (easiest) and Ramp (hardest), under full RGB observation Fig. 1 versus without visual cones Fig. 3, we observe that policies trained *with* visual cones achieve higher success rates on evaluation sets. This suggests visual cones provide a strong, scene-agnostic visual pattern that helps the policy focus on goal-relevant features and improves generalization to unseen scenes.

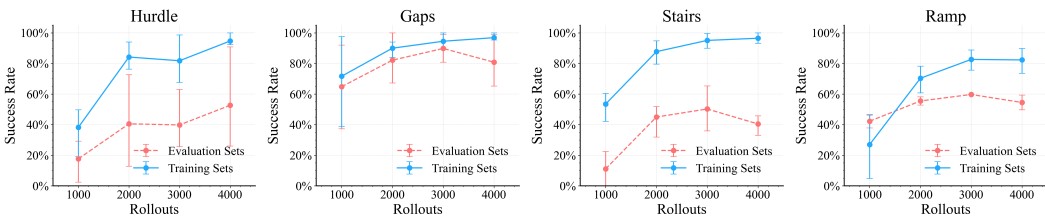

Figure 3: **Task-Specific Multi-Scene Closed-Loop Training. Observation Type: RGB, w/o Visual Cones.**

**Effect of Directional Information.** We also study the impact of explicit direction input in the observation space. Comparing policies trained with direction cues Fig. 4-(A) and without them Fig. 4-(C), we find a moderate drop in evaluation performance when direction is removed. This suggests that while directional input improves performance, policies can still be trained to rely primarily on visual signals (e.g. visual cones) for navigation, albeit with some trade-off in effectiveness.

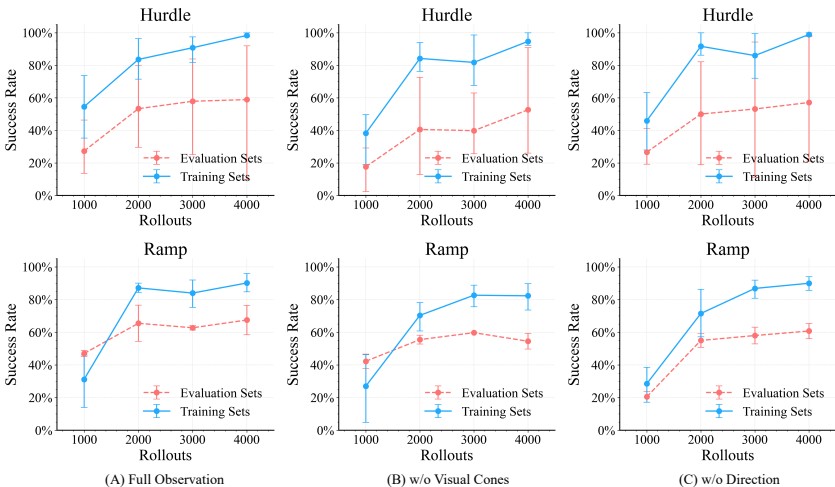

Figure 4: **Ablation on Different Observation Cues**

## 2 Scene Labeling Workflow

The Neverwhere toolchain automates the creation of high-fidelity physical digital twins from uncalibrated images and videos. However, due to random pose initialization, the resulting scenes often have arbitrary orientation and scale. Since robotics applications require a consistent frame of reference, especially to define gravity, some manual labeling is necessary. Thus, we designed an efficient labeling tool by integrating our annotation system with the visualization platform Vuer [2]. This streamlined design allows annotators to label a scene in approximately **one minute**, significantly improving the overall workflow efficiency. The full process, illustrated in Fig. 5, consists of the following steps:

(1) Load the unprocessed collision geometry into the labeling system.
(2) Manually rotate the geometry to align with the Z-up orientation (~20 seconds).
(3) Place two markers on the mesh and input the real-world distance between them; the system automatically computes and applies the scale factor (~10 seconds).
(4) The mesh is automatically cropped, no human input required.
(5) Define waypoints for specific locomotion tasks (~15 seconds).
(6) Click "Save" to automatically generate and export the scene's XML configuration.

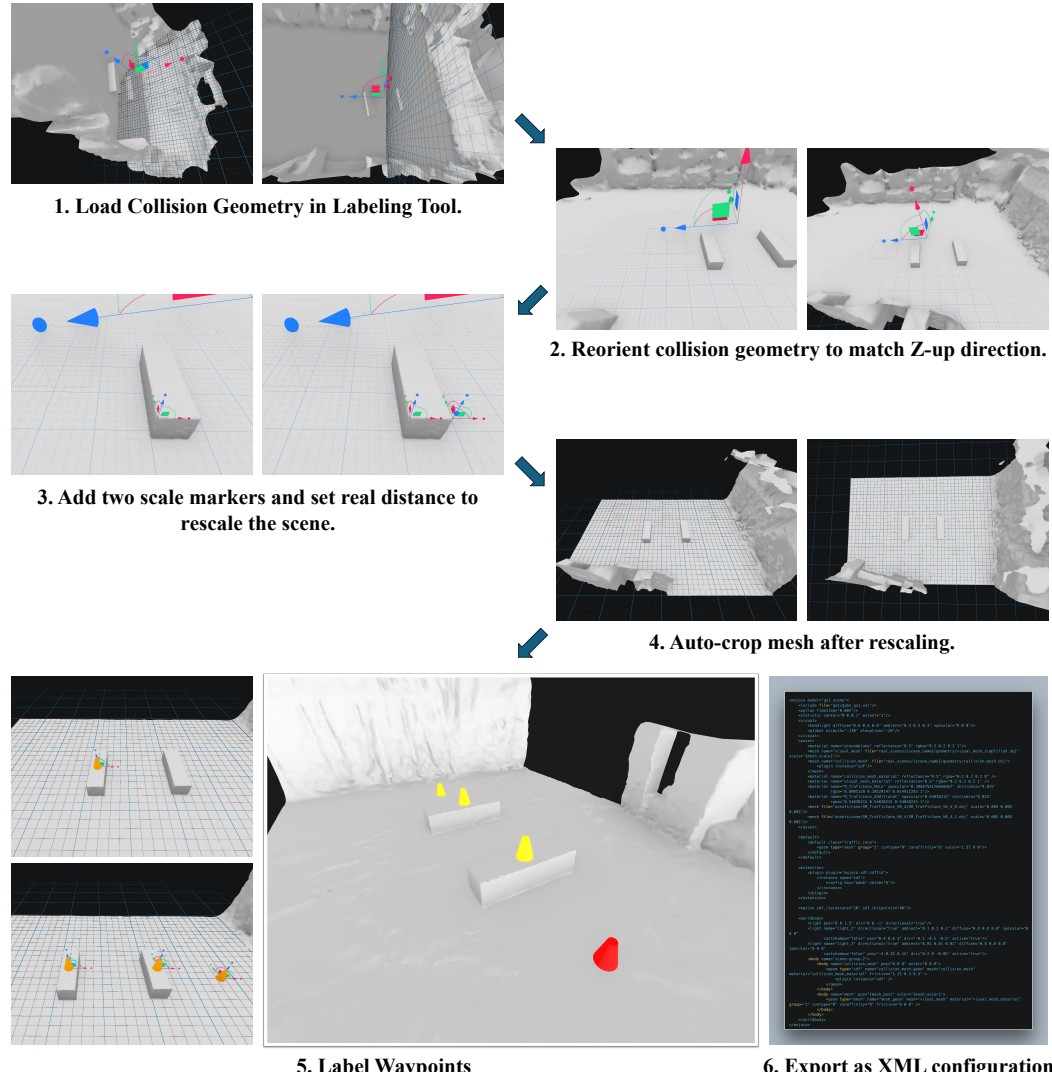

**1. Load Collision Geometry in Labeling Tool.**

**2. Reorient collision geometry to match Z-up direction.**

**3. Add two scale markers and set real distance to rescale the scene.**

**4. Auto-crop mesh after rescaling.**

**5. Label Waypoints**

**6. Export as XML configuration**

Figure 5: **Overview of Scene Labeling Workflow.**

## 3 Gallery of Neverwhere Scenes and Assets

We provide over 60 high-quality, ready-to-use scenes in the NeverWhere benchmark. Below, we showcase all of them. Each figure displays, from left to right: **(1) the original scan, (2) a 3DGS [1] rendering from a viewpoint similar to (1), and (3) an overview of the full 3DGS scene.** All scenes can be accessed via link. For details on the implementation of the scene creation tool, please visit our anonymous project website for code.

Note that the scenes listed here are referred to by their original scan names. For the closed-loop training and evaluation experiments described in the main paper, we use descriptive scene names to better reflect each scene's characteristics. A mapping between the original scan names and the descriptive names is provided in Tab. 1.

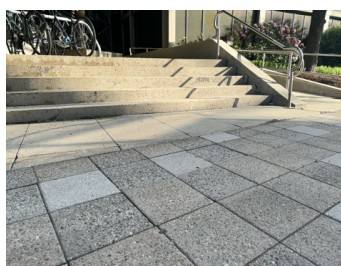 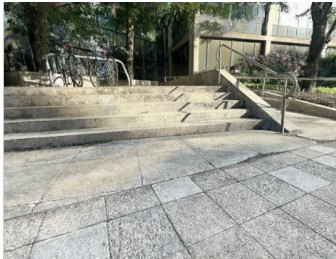 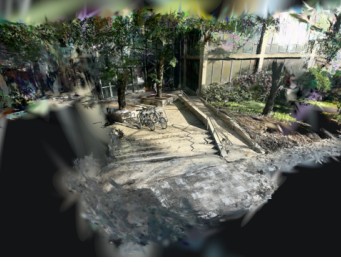

Figure 6: Scene name: **building_31_stairs_v1**; Scene type: **Stairs**.

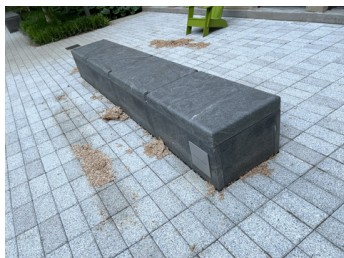 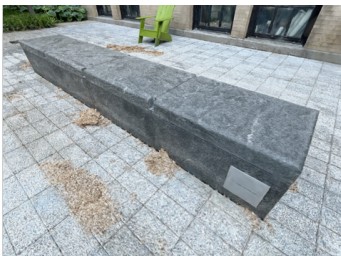 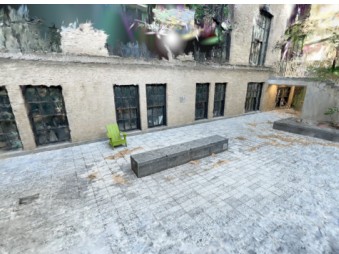

Figure 7: Scene name: **hurdle_black_stone_v1**; Scene type: **Hurdle**.

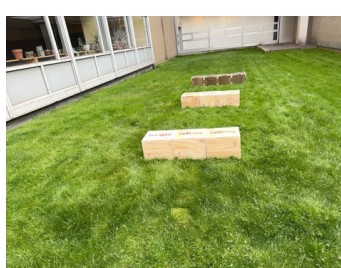 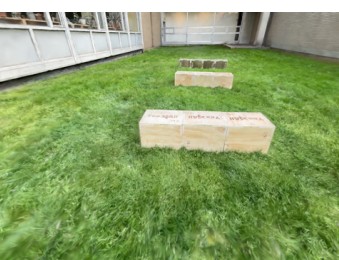 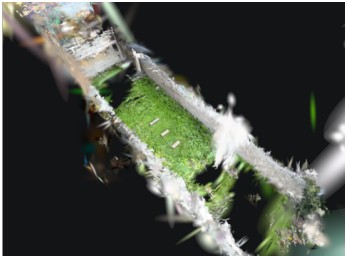

Figure 8: Scene name: **hurdle_three_grassy_courtyard_v2**; Scene type: **Hurdle**.

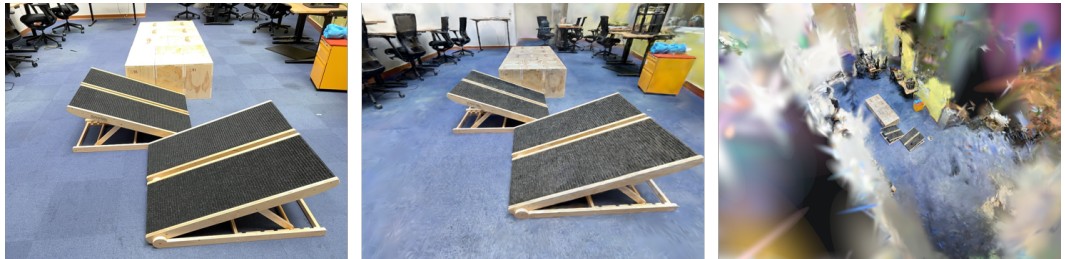

Figure 9: Scene name: **ramp_spread_blue_carpet_v5**; Scene type: **Ramp**.

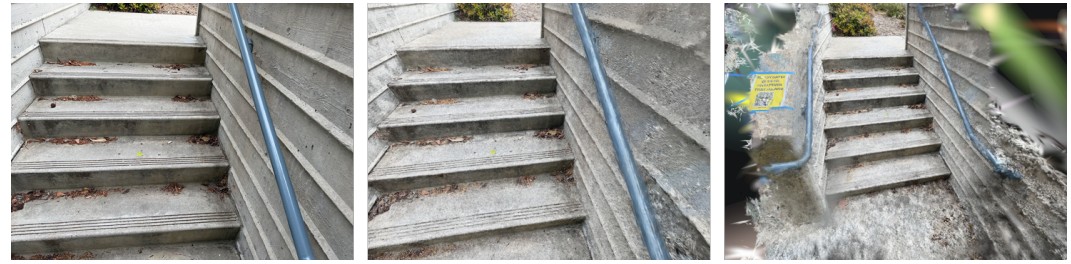

Figure 10: Scene name: **stairs_jacobs_front**; Scene type: **Stairs**.

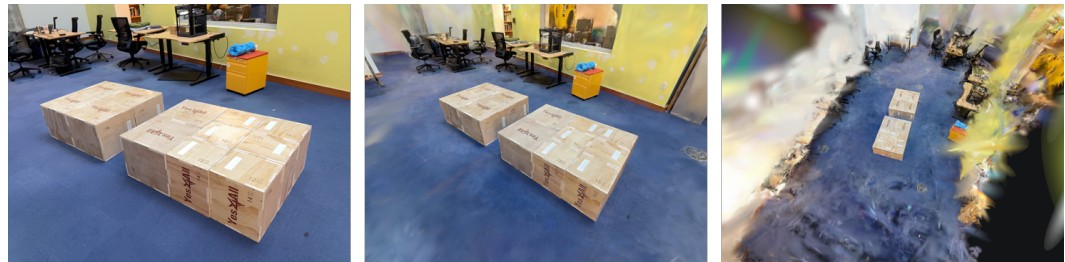

Figure 11: Scene name: **gaps_12in_226_blue_carpet_v2**; Scene type: **Gaps**.

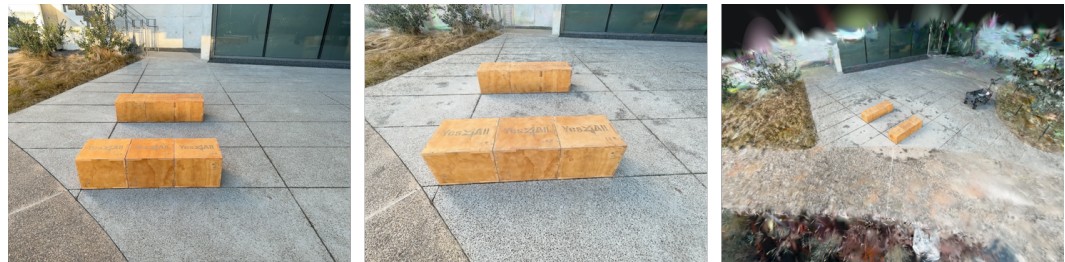

Figure 12: Scene name: **hurdle_fah_back_two_hurldes_wood_v1**; Scene type: **Hurdle**.

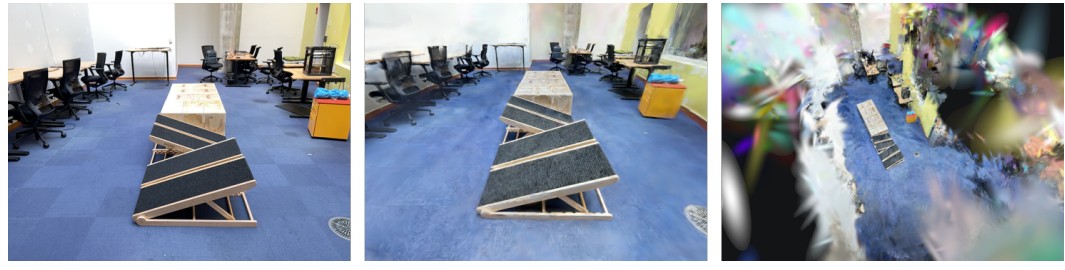

Figure 13: Scene name: **ramp_aligned_blue_carpet_v4**; Scene type: **Ramp**.

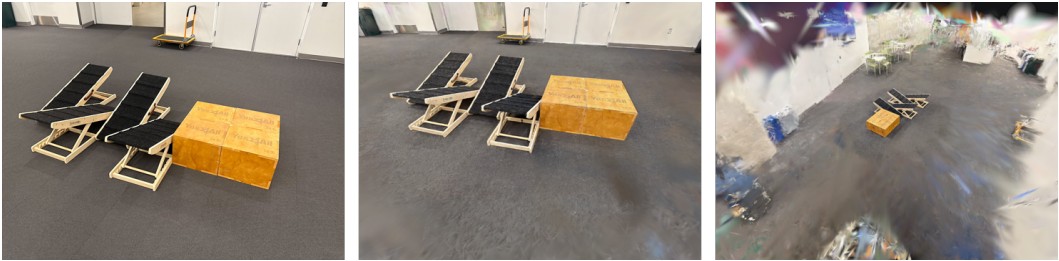

Figure 14: Scene name: **ramps_fah_ll_indoor**; Scene type: **Ramp**.

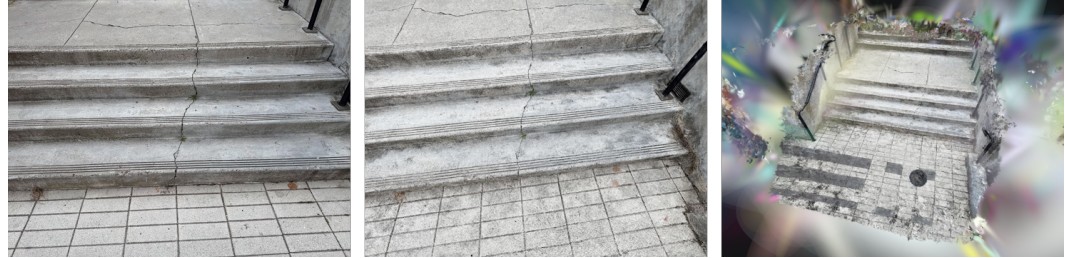

Figure 15: Scene name: **stairs_pc_fountain**; Scene type: **Stairs**.

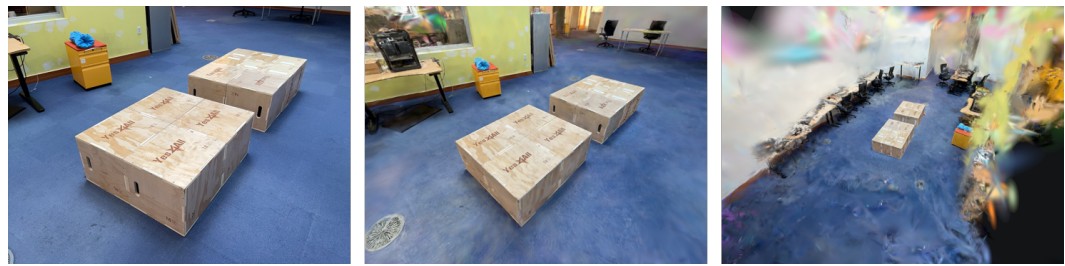

Figure 16: Scene name: **gaps_16in_226_blue_carpet_v2**; Scene type: **Gaps**.

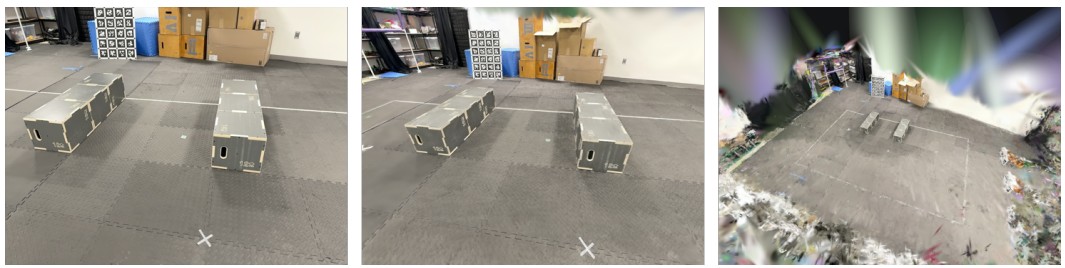

Figure 17: Scene name: **hurdle_fah_indoor_two_hurdle_rubber_v1**; Scene type: **Hurdle**.

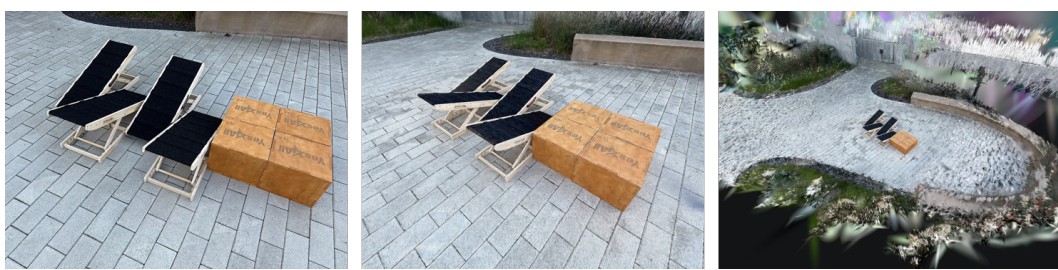

Figure 18: Scene name: **ramp_atkinson_back**; Scene type: **Ramp**.

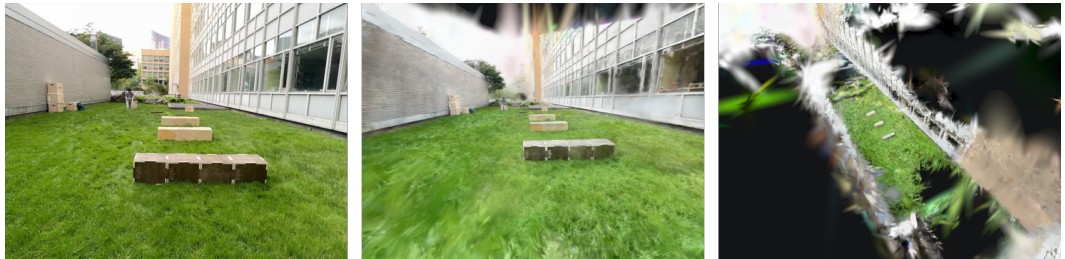

Figure 19: Scene name: **real_hurdle_three_grassy_ally_v2**; Scene type: **Hurdle**.

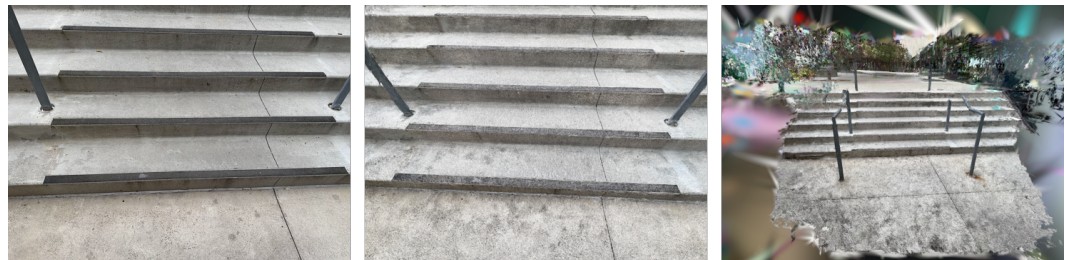

Figure 20: Scene name: **stairs_pc_side**; Scene type: **Stairs**.

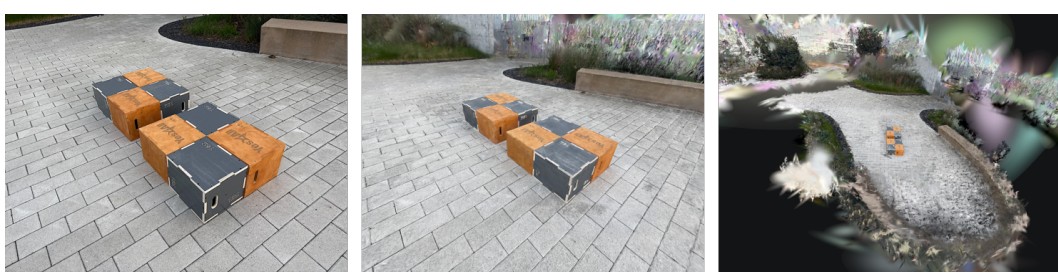

Figure 21: Scene name: **gaps_atkinson_back_12in**; Scene type: **Gaps**.

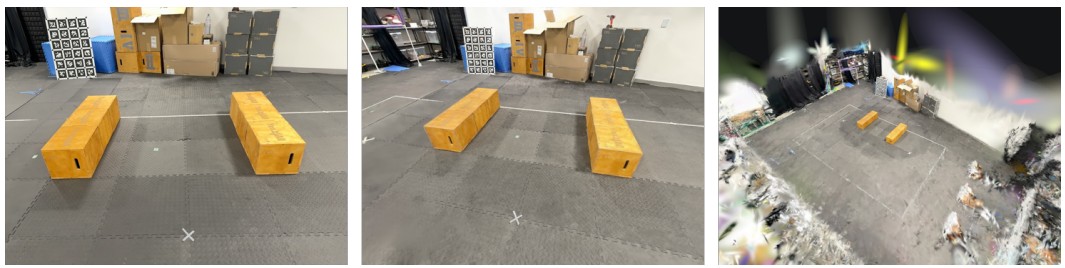

Figure 22: Scene name: **hurdle_fah_indoor_two_hurdle_wood_v1**; Scene type: **Hurdle**.

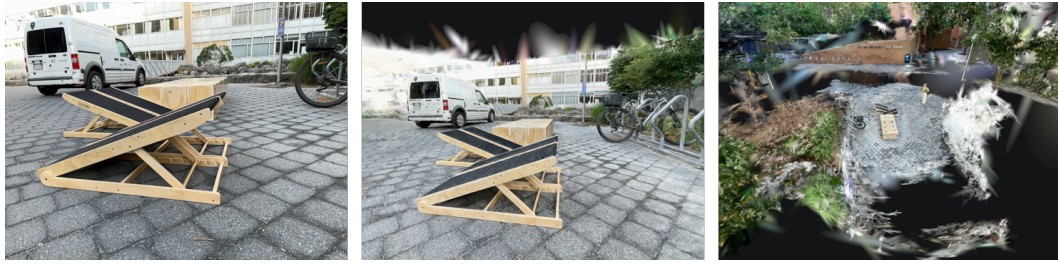

Figure 23: Scene name: **ramp_bricks_v2**; Scene type: **Ramp**.

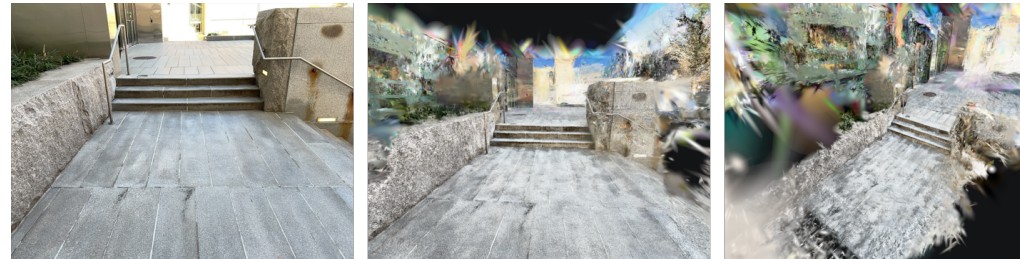

Figure 24: Scene name: **real_stair_02_bcs_v1**; Scene type: **Stairs**.

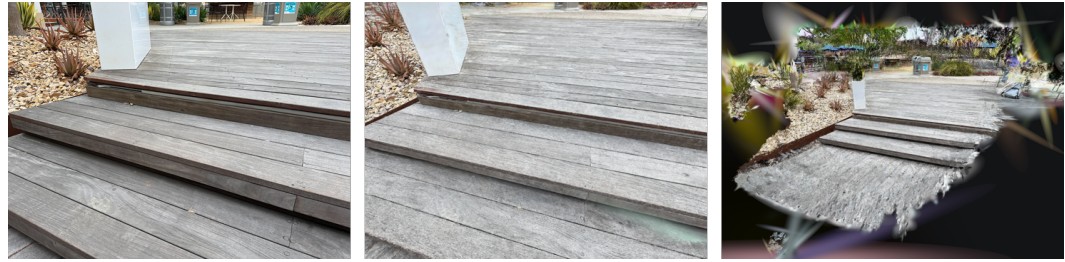

Figure 25: Scene name: **stairs_pcw**; Scene type: **Stairs**.

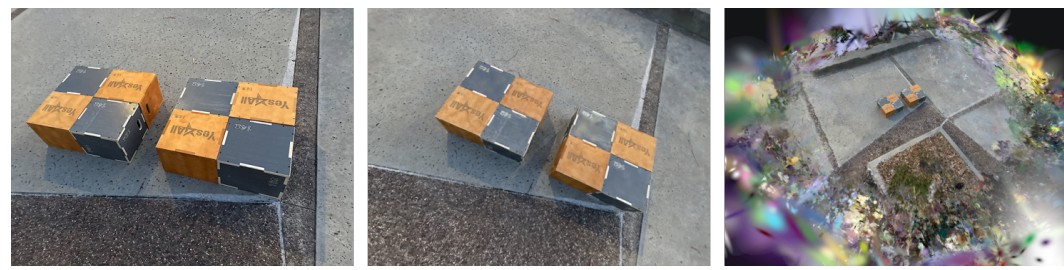

Figure 26: Scene name: **gaps_center_12in**; Scene type: **Gaps**.

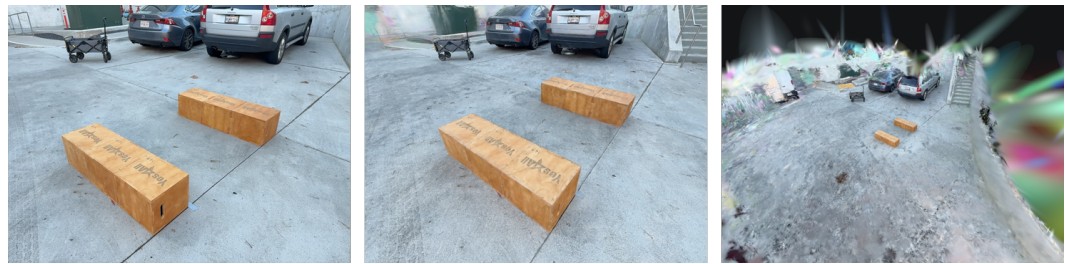

Figure 27: Scene name: **hurdle_fah_ll_two_hurldes_wood_v1**; Scene type: **Hurdle**.

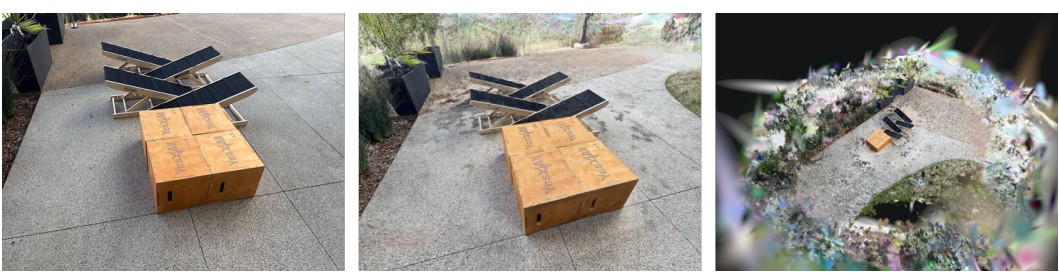

Figure 28: Scene name: **ramp_fah_back**; Scene type: **Ramp**.

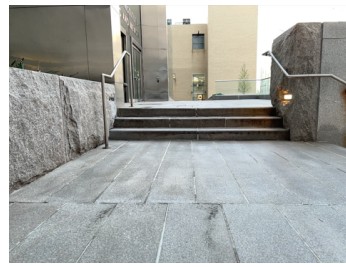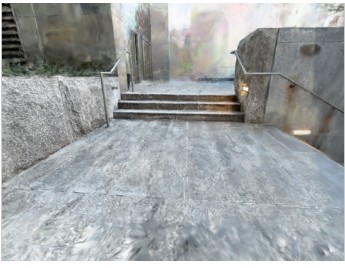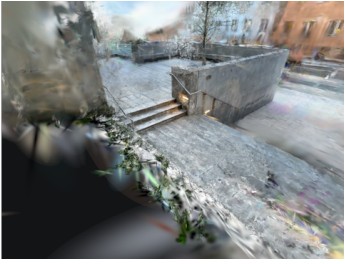

Figure 29: Scene name: **real_stair_04_bcs_dusk**; Scene type: **Stairs**.

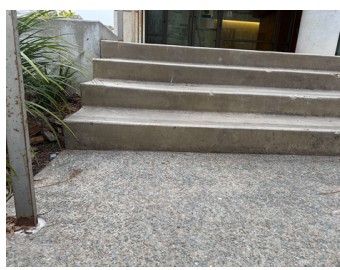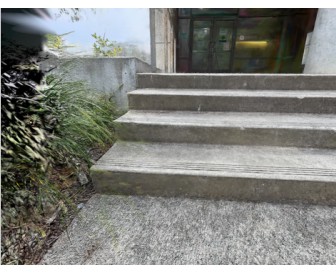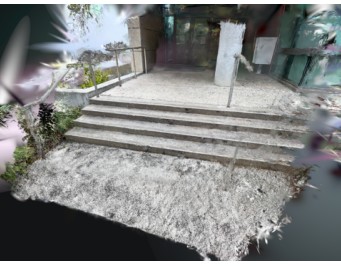

Figure 30: Scene name: **stairs_pfbh_front**; Scene type: **Stairs**.

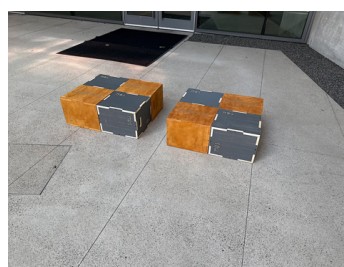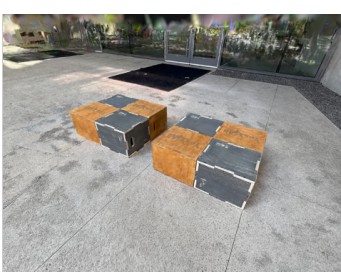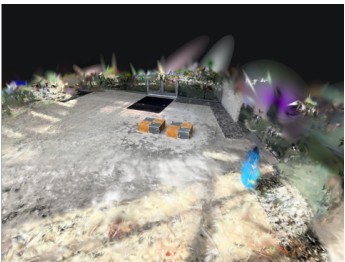

Figure 31: Scene name: **gaps_fah_front_12in**; Scene type: **Gaps**.

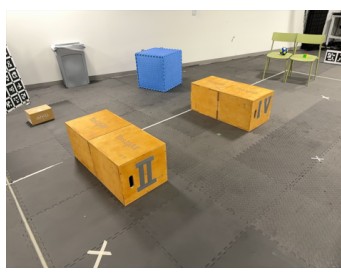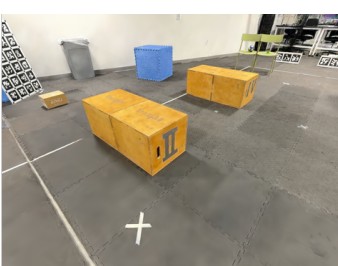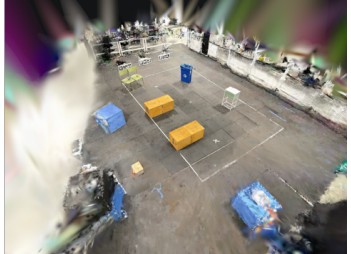

Figure 32: Scene name: **hurdle_fah_two_hurdle_gray_ground_v1**; Scene type: **Hurdle**.

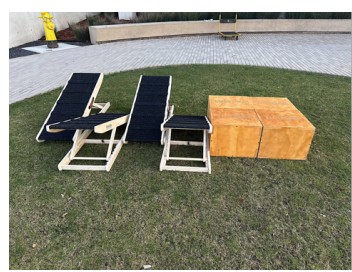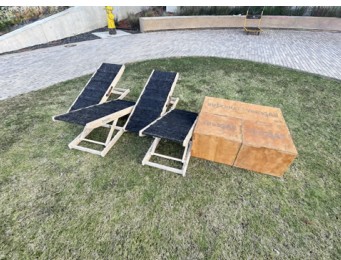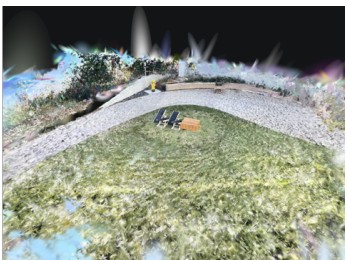

Figure 33: Scene name: **ramp_fah_garden**; Scene type: **Ramp**.

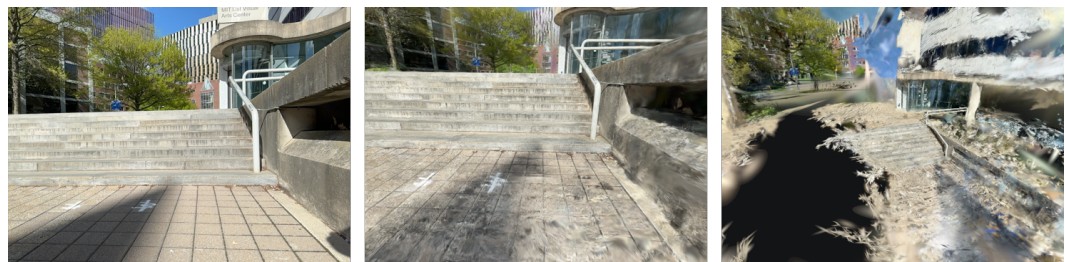

Figure 34: Scene name: **real_stair_08_mc_afternoon_v1**; Scene type: **Stairs**.

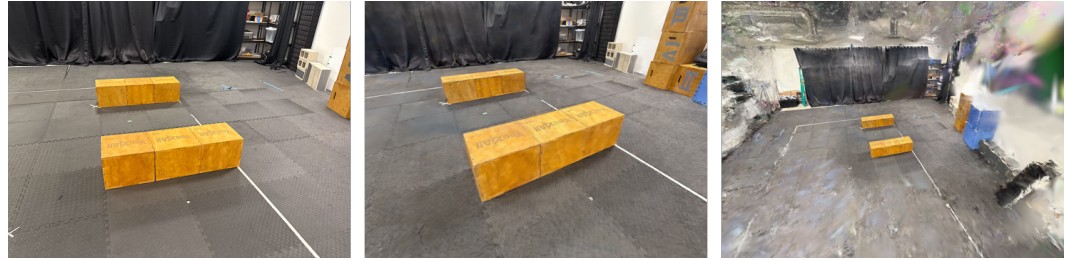

Figure 35: Scene name: **test_real_robot_sample_1**; Scene type: **Test Sample**.

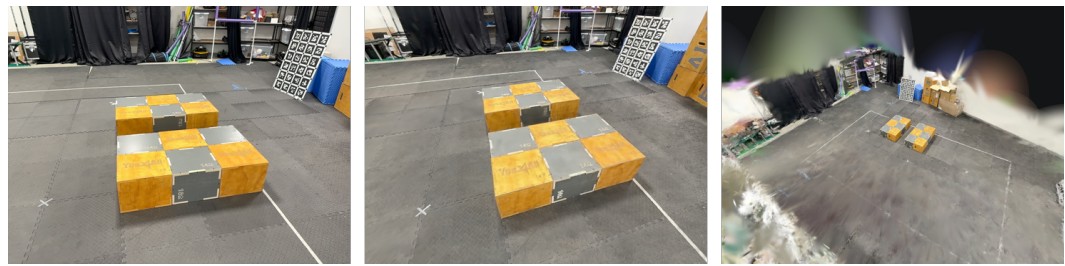

Figure 36: Scene name: **gaps_fah_ll_gaps_15in_indoor_mixtex_v1**; Scene type: **Gaps**.

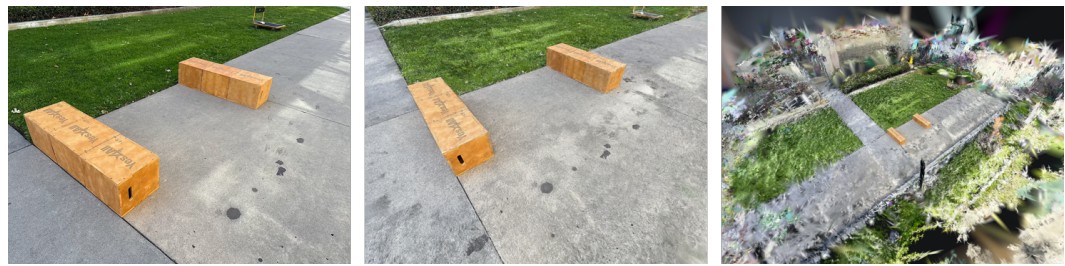

Figure 37: Scene name: **hurdle_jacobs_side**; Scene type: **Hurdle**.

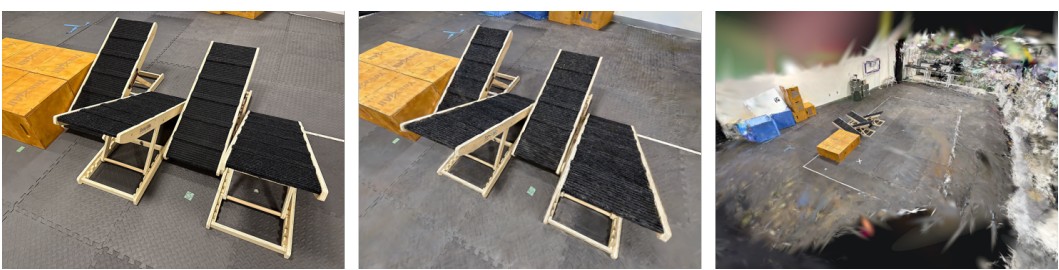

Figure 38: Scene name: **ramp_fah_indoor_wood_4_ramp_v1**; Scene type: **Ramp**.

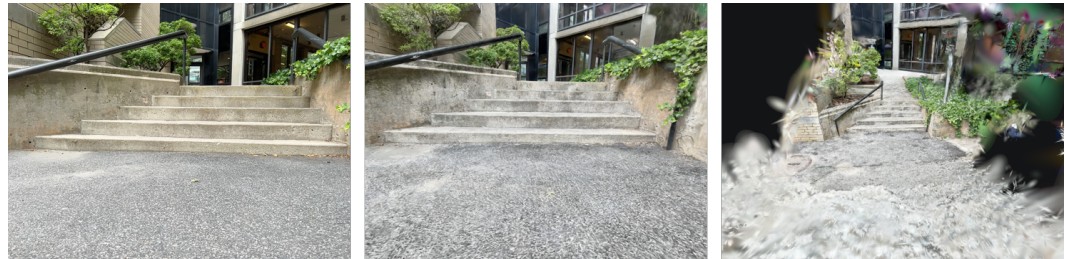

Figure 39: Scene name: **stairs_36_backstairs_v2**; Scene type: **Stairs**.

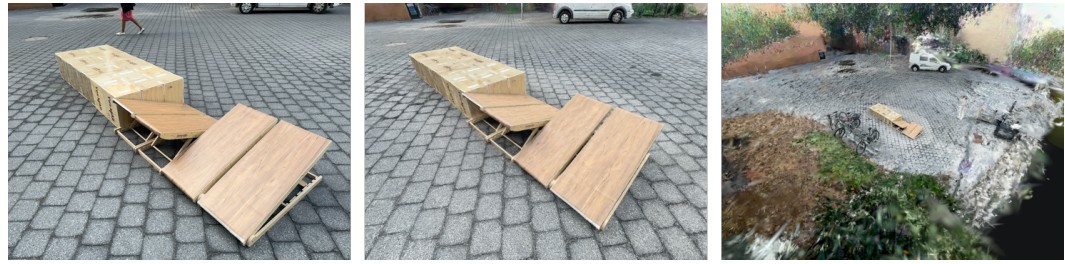

Figure 40: Scene name: **wood_ramp_aligned_bricks_v1**; Scene type: **Ramp**.

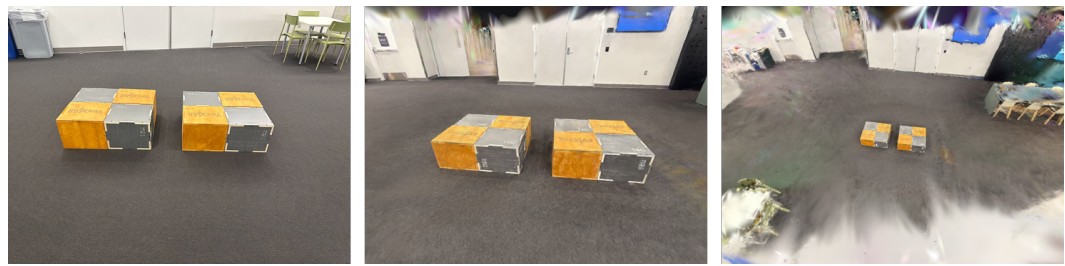

Figure 41: Scene name: **gaps_fah_ll_indoor_12in**; Scene type: **Gaps**.

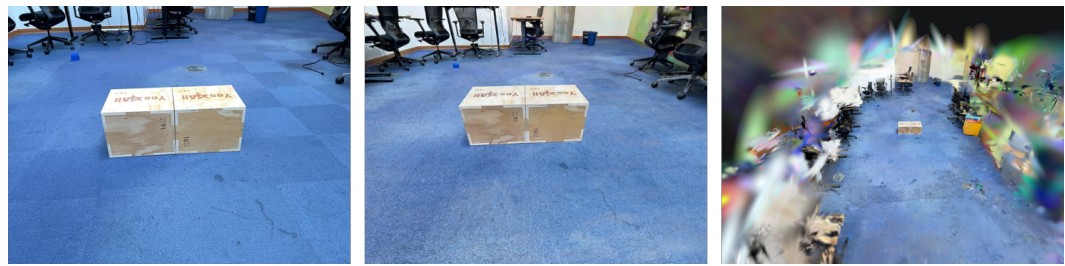

Figure 42: Scene name: **hurdle_one_blue_carpet_v2**; Scene type: **Hurdle**.

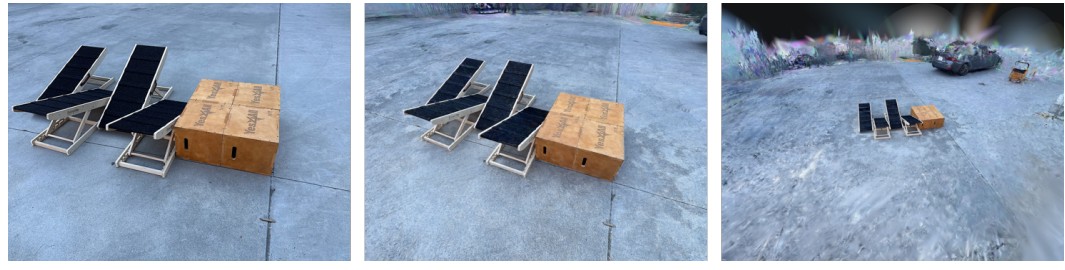

Figure 43: Scene name: **ramp_fah_side**; Scene type: **Ramp**.

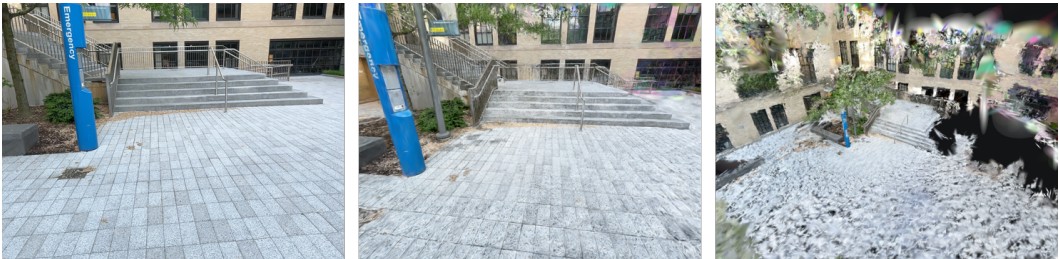

Figure 44: Scene name: **stairs_4_stairs2up_v1**; Scene type: **Stairs**.

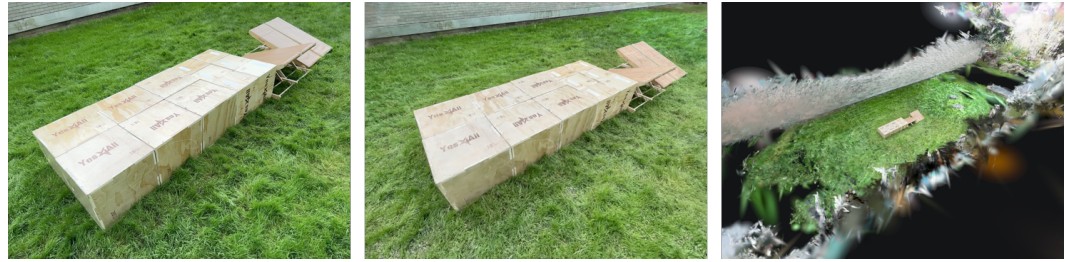

Figure 45: Scene name: **wood_ramp_aligned_grass_v2**; Scene type: **Ramp**.

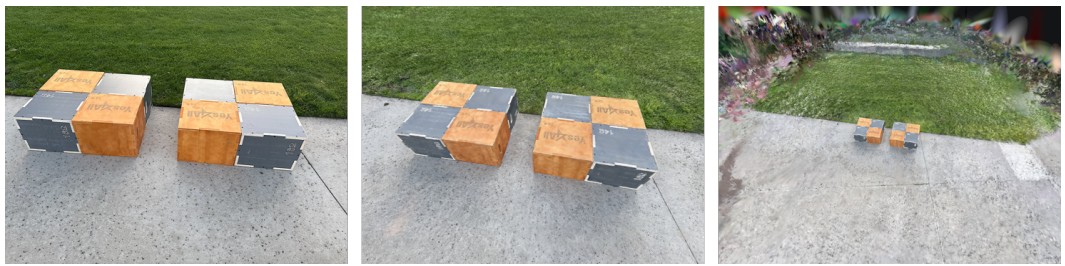

Figure 46: Scene name: **gaps_geisel_12in**; Scene type: **Gaps**.

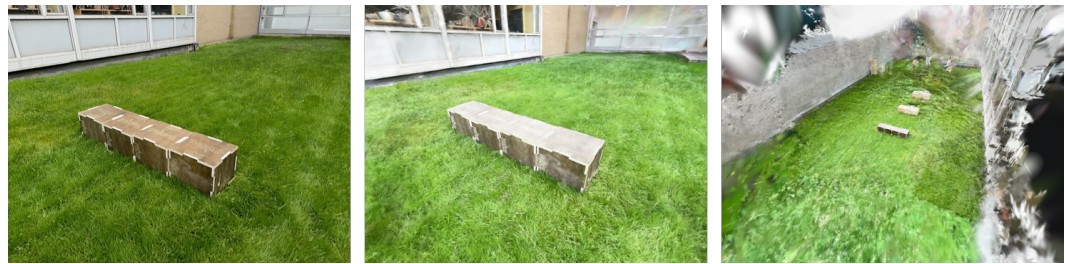

Figure 47: Scene name: **hurdle_one_dark_grassy_courtyard_v1**; Scene type: **Hurdle**.

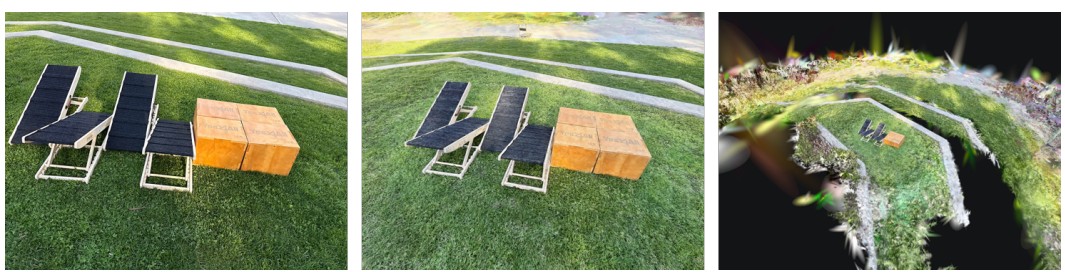

Figure 48: Scene name: **ramp_geisel**; Scene type: **Ramp**.

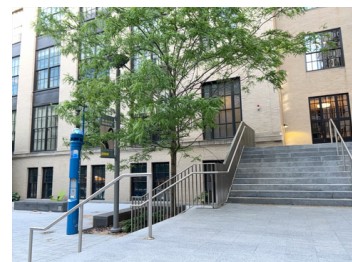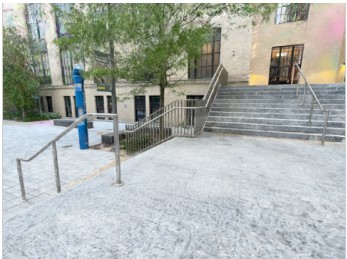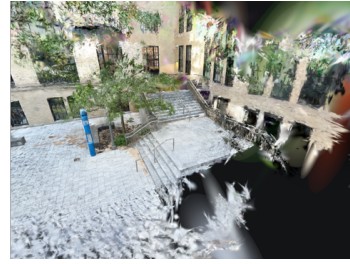

Figure 49: Scene name: **stairs_48_v3**; Scene type: **Stairs**.

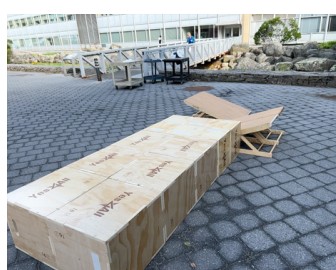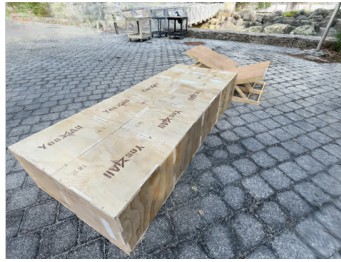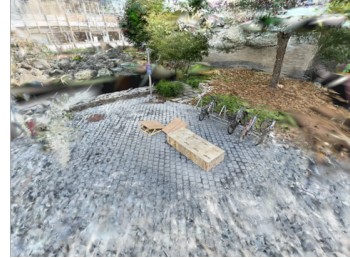

Figure 50: Scene name: **wood_ramp_offset_bricks_v2**; Scene type: **Ramp**.

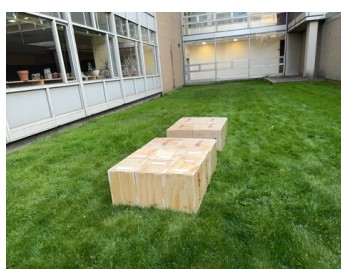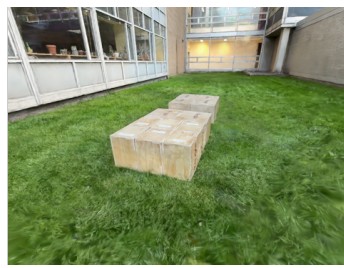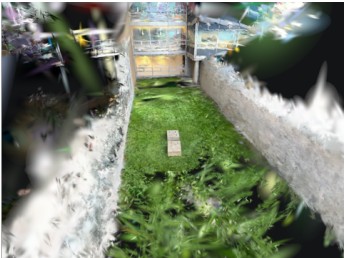

Figure 51: Scene name: **gaps_grassy_courtyard_v2**; Scene type: **Gaps**.

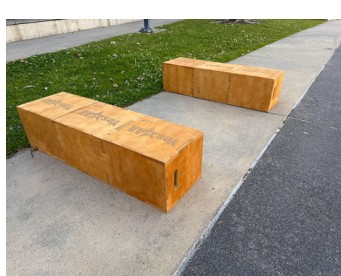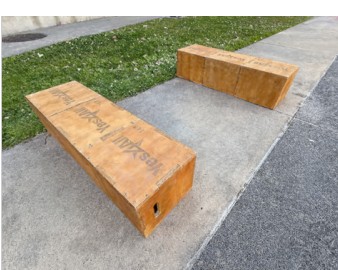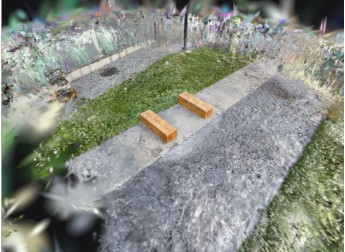

Figure 52: Scene name: **hurdle_pfb_side**; Scene type: **Hurdle**.

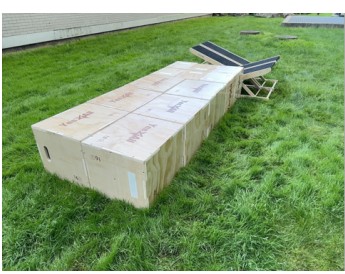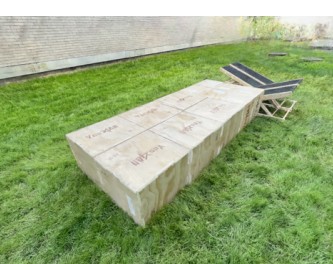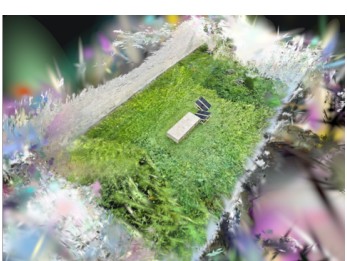

Figure 53: Scene name: **ramp_grass_v3**; Scene type: **Ramp**.

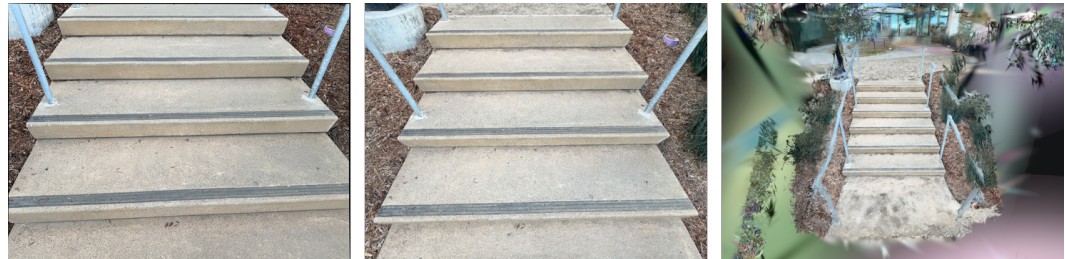

Figure 54: Scene name: **stairs_atkinson_back**; Scene type: **Stairs**.

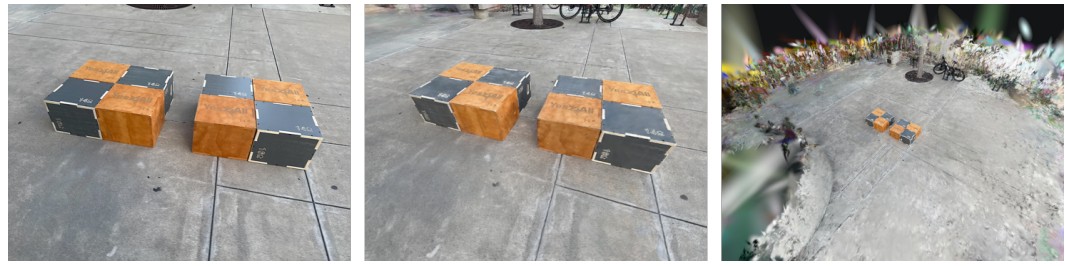

Figure 55: Scene name: **gaps_jacobs_front_12in**; Scene type: **Gaps**.

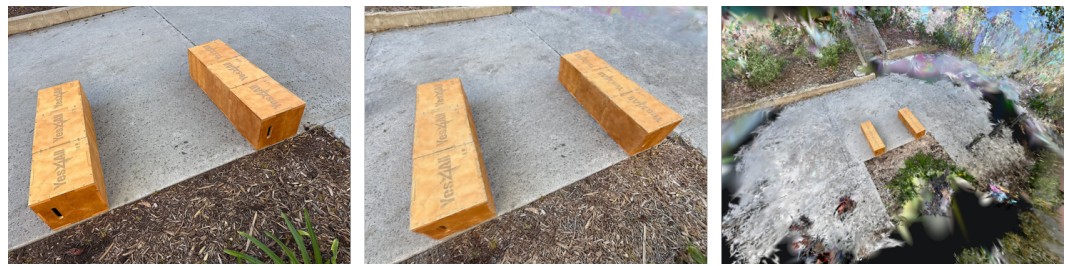

Figure 56: Scene name: **hurdle_pssl_side**; Scene type: **Hurdle**.

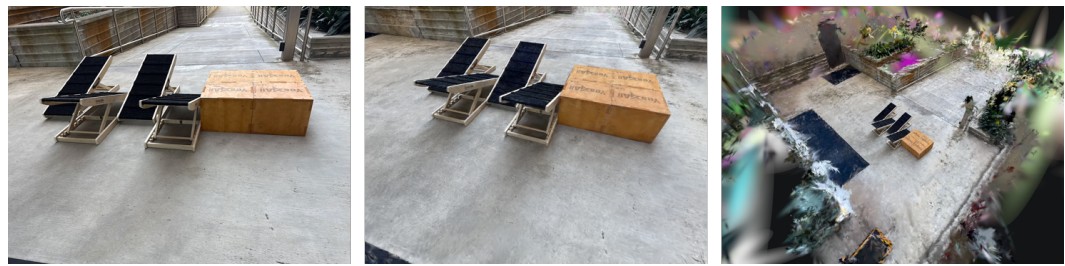

Figure 57: Scene name: **ramp_jacobs_back**; Scene type: **Ramp**.

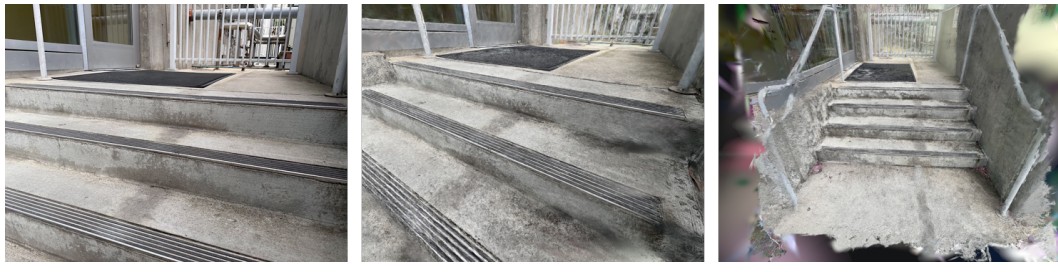

Figure 58: Scene name: **stairs_atkinson_side**; Scene type: **Stairs**.

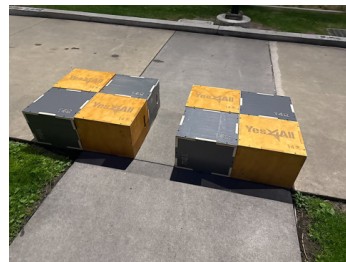 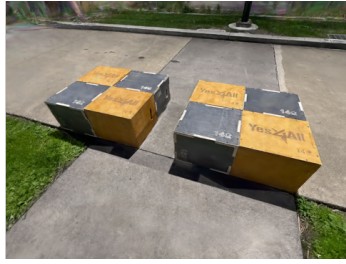 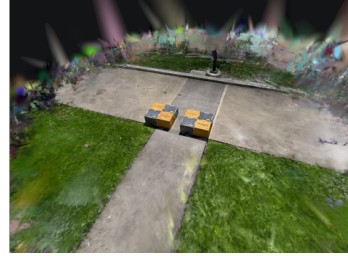

Figure 59: Scene name: **gaps_jacobs_side_12in**; Scene type: **Gaps**.

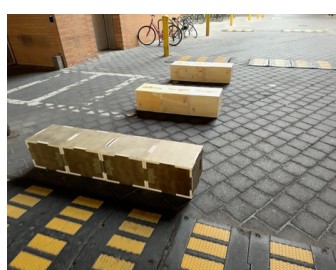 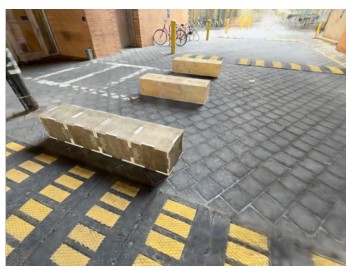 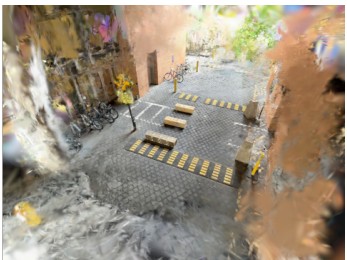

Figure 60: Scene name: **hurdle_stata_v1**; Scene type: **Hurdle**.

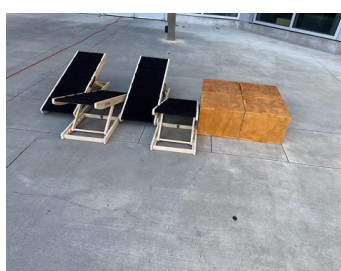 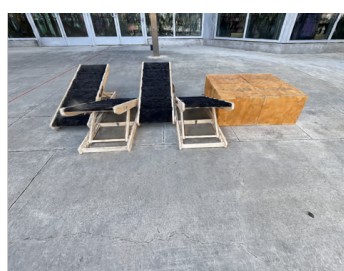 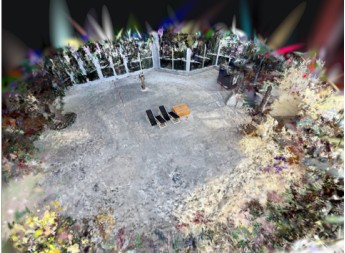

Figure 61: Scene name: **ramp_jacobs_front**; Scene type: **Ramp**.

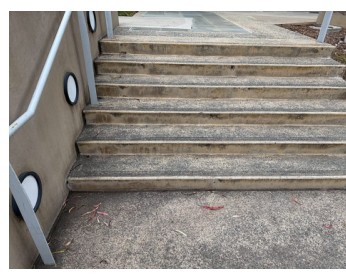 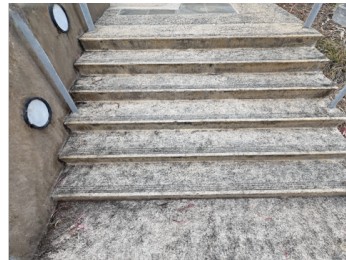 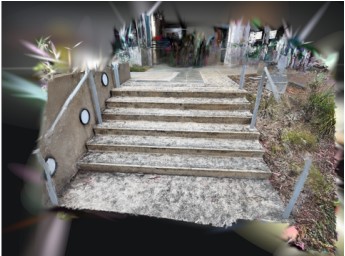

Figure 62: Scene name: **stairs_ewc_front**; Scene type: **Stairs**.

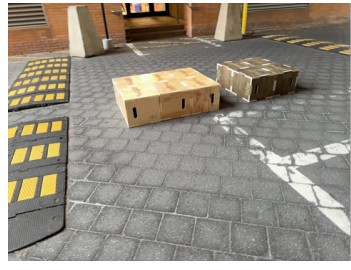 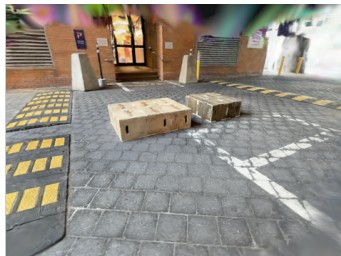 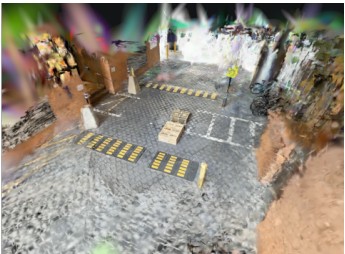

Figure 63: Scene name: **gaps_stata_v1**; Scene type: **Gaps**.

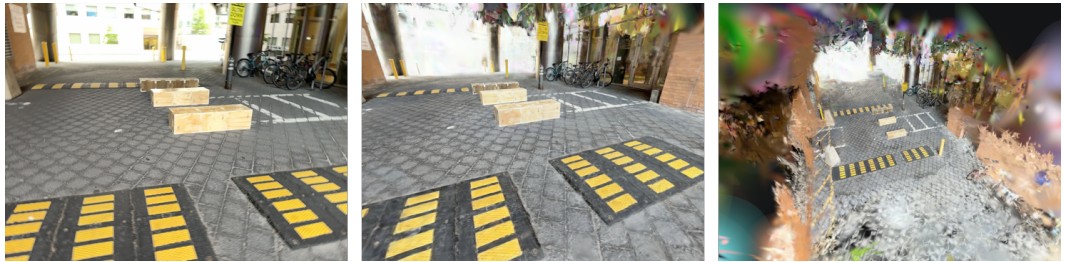

Figure 64: Scene name: **hurdle_stata_v2**; Scene type: **Hurdle**.

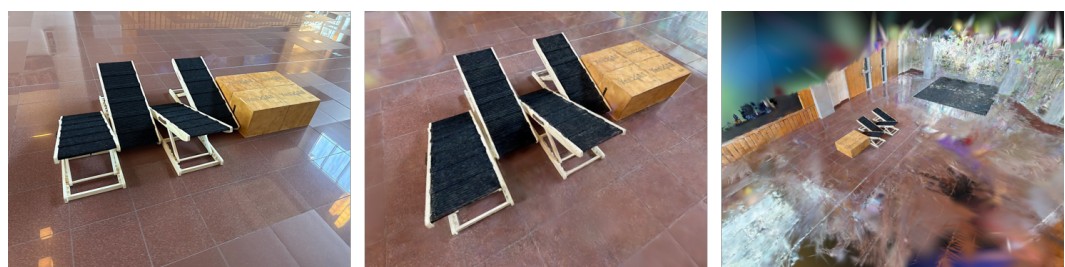

Figure 65: Scene name: **ramp_jacobs_indoor**; Scene type: **Ramp**.

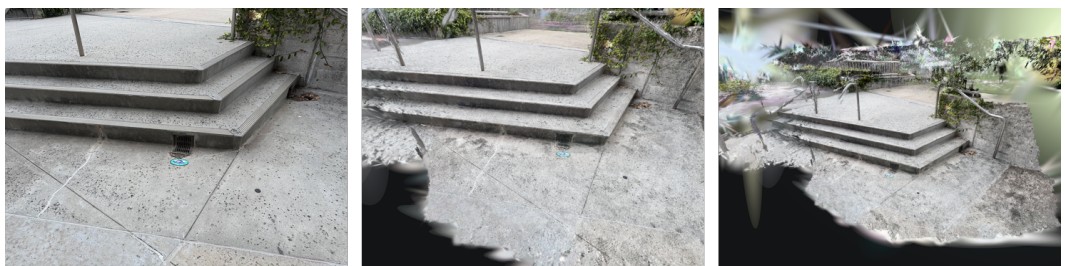

Figure 66: Scene name: **stairs_geisel_up**; Scene type: **Stairs**.

Table 1: Mapping from Scan Names to Descriptive Names.

| Scan Name | Descriptive Name |
|---|---|
| gaps_12in_226_blue_carpet_v2 | Indoor A Carpet 0 |
| gaps_16in_226_blue_carpet_v2 | Indoor A Carpet 1 |
| gaps_grassy_courtyard_v2 | Outdoor A Grass 0 |
| gaps_stata_v1 | Outdoor A Cobblestone 0 |
| gaps_atkinson_back_12in | Outdoor B Cobblestone 0 |
| gaps_center_12in | Outdoor B Concrete 0 |
| gaps_fah_front_12in | Outdoor B Concrete 1 |
| gaps_fah_ll_gaps_15in_indoor_mixtex_v1 | Indoor B Foam_tiles 0 |
| gaps_fah_ll_indoor_12in | Indoor B Carpet 0 |
| gaps_geisel_12in | Outdoor B Concrete&Grass 0 |
| gaps_jacobs_front_12in | Outdoor B Concrete 2 |
| gaps_jacobs_side_12in | Outdoor B Concrete&Grass 1 |
| hurdle_226_blue_carpet_v3 | Indoor A Carpet 0 |
| hurdle_black_stone_v1 | Outdoor A Cobblestone 0 |
| hurdle_one_blue_carpet_v2 | Indoor A Carpet 1 |
| hurdle_one_dark_grassy_courtyard_v1 | Outdoor A Grass 0 |
| hurdle_stata_v1 | Outdoor A Cobblestone 1 |
| hurdle_stata_v2 | Outdoor A Cobblestone 2 |
| real_hurdle_three_grassy_ally_v2 | Outdoor A Grass 1 |
| hurdle_fah_back_two_hurldes_wood_v1 | Outdoor B Concrete 0 |
| hurdle_fah_indoor_two_hurdle_wood_v1 | Indoor B Foam_tiles 0 |
| hurdle_fah_indoor_two_hurdle_rubber_v1 | Indoor B Foam_tiles 1 |
| hurdle_fah_ll_two_hurldes_wood_v1 | Outdoor B Concrete 1 |
| hurdle_jacobs_side | Outdoor B Concrete&Grass 0 |
| hurdle_pfb_side | Outdoor B Concrete&Grass 1 |
| hurdle_pssl_side | Outdoor B Concrete 2 |
| test_real_robot_sample_1 | Indoor B Foam_tiles 2 |
| ramp_bricks_v2 | Outdoor A Cobblestone 0 |
| ramp_grass_v3 | Outdoor A Grass 0 |
| wood_ramp_aligned_grass_v2 | Outdoor A Grass 1 |
| ramp_geisel | Outdoor B Grass 0 |
| ramp_jacobs_indoor | Indoor B Ceramic 0 |
| building_31_stairs_v1 | Outdoor A Concrete 0 |
| real_stair_02_bcs_v1 | Outdoor A Bricks 0 |
| real_stair_04_bcs_dusk | Outdoor A Bricks 1 |
| real_stair_08_mc_afternoon_v1 | Outdoor A Concrete 1 |
| stairs_36_backstairs_v2 | Outdoor A Concrete 2 |
| stairs_48_v3 | Outdoor A Bricks 2 |
| stairs_4_stairs2up_v1 | Outdoor A Bricks 3 |
| stairs_atkinson_back | Outdoor B Concrete 0 |
| stairs_atkinson_side | Outdoor B Concrete 1 |
| stairs_ewc_front | Outdoor B Concrete 2 |
| stairs_fah_back | Outdoor B Concrete 3 |
| stairs_geisel_up | Outdoor B Concrete 4 |
| stairs_pcw | Outdoor B Wood 0 |
| stairs_pfbh_front | Outdoor B Concrete 5 |

| Task Type | Scene Name |
| --- | --- |
| Gaps | gaps_fah_ll_gaps_15in_indoor_mixtex_v1 |
| Gaps | gaps_center_12in |
| Gaps | gaps_grassy_courtyard_v2 |
| Gaps | gaps_fah_ll_indoor_12in |
| Gaps | gaps_geisel_12in |
| Gaps | gaps_fah_front_12in |
| Gaps | gaps_jacobs_side_12in |
| Gaps | gaps_stata_v1 |
| Gaps | gaps_atkinson_back_12in |
| Gaps | gaps_12in_226_blue_carpet_v2 |
| Gaps | gaps_16in_226_blue_carpet_v2 |
| Gaps | gaps_jacobs_front_12in |
| Hurdle | hurdle_one_blue_carpet_v2 |
| Hurdle | hurdle_pfb_side |
| Hurdle | hurdle_fah_indoor_two_hurdle_wood_v1 |
| Hurdle | hurdle_fah_back_two_hurldes_wood_v1 |
| Hurdle | hurdle_stata_v1 |
| Hurdle | real_hurdle_three_grassy_ally_v2 |
| Hurdle | hurdle_stata_v2 |
| Hurdle | hurdle_fah_ll_two_hurldes_wood_v1 |
| Hurdle | hurdle_pssl_side |
| Hurdle | hurdle_fah_indoor_two_hurdle_rubber_v1 |
| Hurdle | hurdle_jacobs_side |
| Hurdle | hurdle_one_dark_grassy_courtyard_v1 |
| Hurdle | hurdle_black_stone_v1 |
| Hurdle | test_real_robot_sample_1 |
| Hurdle | hurdle_226_blue_carpet_v3 |
| Stairs | stairs_atkinson_back |
| Stairs | stairs_atkinson_side |
| Stairs | real_stair_08_mc_afternoon_v1 |
| Stairs | stairs_geisel_up |
| Stairs | stairs_fah_back |
| Stairs | stairs_pfbh_front |
| Stairs | real_stair_02_bcs_v1 |
| Stairs | stairs_pcw |
| Stairs | stairs_48_v3 |
| Stairs | stairs_4_stairs2up_v1 |
| Stairs | real_stair_04_bcs_dusk |
| Stairs | building_31_stairs_v1 |
| Stairs | stairs_36_backstairs_v2 |
| Stairs | stairs_ewc_front |
| Ramp | ramp_jacobs_indoor |
| Ramp | ramp_geisel |
| Ramp | ramp_grass_v3 |
| Ramp | wood_ramp_aligned_grass_v2 |
| Ramp | ramp_bricks_v2 |

Table 2: Scenes used in multi-scene closed-loop training. Training set scenes are colored light cyan, and evaluation set scenes are colored light orange.