# OpenReview forum: "The Neverwhere Visual Parkour Benchmark Suite"
_NeurIPS.cc/2025/Datasets_and_Benchmarks_Track — Submitted to NeurIPS 2025 Datasets and Benchmarks Track_

### Official Review · Reviewer_kab6 · 2025-06-30

**Rating:** 4
**Confidence:** 3

**Summary:**

This research presents the Neverwhere Benchmark Suite and its accompanying toolchain, aiming to solve the challenge of robot policy evaluation. Its primary contributions are: 1) providing a series of high-visual-fidelity evaluation environments built with 3D Gaussian Splatting technology; and 2) developing a toolchain that can automatically generate visual models and accurate physical collision meshes from photos. Experiments demonstrate that the benchmark can effectively measure a policy's generalization ability, positioning it as a deterministic evaluation tool rather than a diverse data source for training.

**Dataset Code Accessibility:**

Yes

**Dataset Code Comments:**

The paper provides a link to an anonymous website for direct access to the code and dataset.

**Ethical Comments:**

No significant ethical concerns were identified with this work. The authors' handling of the NeurIPS checklist is thorough and reasonable.

The research is foundational, providing a technical benchmark for robot evaluation rather than a deployed application, which justifies their assessment that a broader societal impact discussion is not directly applicable. The data is sourced from university campuses for geometric simulation and does not involve human subjects as study participants, minimizing privacy risks and the need for special safeguards.

**Ethical Considerations:**

No, there are no or only very minor ethics concerns

**Limitations Weaknesses:**

1. **Manual Annotation Limits Scalability:** A key weakness, acknowledged by the authors, is that the pipeline requires manual reorientation, rescaling, and task labeling (lines 267-270). This creates a significant bottleneck for scaling the number and diversity of scenes.

2. **Lack of Depth Data Validation:** The benchmark's provided depth data leads to very poor policy performance (Fig. 10-C) , with the authors noting it is "less effective for learning". This is a significant limitation, as depth is a critical modality for locomotion. Other methods, such as in LucidSim, have shown that performance differences between depth and RGB policies are not significant. The extremely poor performance here raises serious questions about the benchmark's depth data quality. The work lacks direct validation (e.g., comparison to ground truth) to prove the data's integrity.

3. **Absence of Dynamic Environments:** All environments are entirely static. This is a major limitation as it prevents the evaluation of crucial capabilities like dynamic obstacle avoidance and interaction. The authors acknowledge this when noting the benchmark "lacks... the ability to animate other actors" (line 71).

**Strengths Contributions:**

1.  **A complete, open-source real-to-sim toolchain.** This pipeline is the core of the work, addressing a key challenge by automatically generating high-fidelity digital twins from real-world photos. It innovatively produces two aligned assets: photorealistic visuals using 3D Gaussian Splatting and a separate, accurate collision mesh for reliable physics simulation.

2.  **A clear and evidence-backed framework for robot policy evaluation.** The authors leverage their technical achievement to position Neverwhere as a deterministic benchmark for testing generalization, distinctly contrasting it with data-diverse tools meant for training. This crucial argument is delivered with exceptional clarity through a well-organized paper and highly informative figures.

---

> ### Author Rebuttal · Authors · 2025-07-31
>
> We thank the reviewer for their insightful feedback, and address key concerns below.
>
> **Modeling dynamic scenes is within our framework's capability**
>
> > All environments are entirely static. This is a major limitation as it prevents the evaluation of crucial capabilities like dynamic obstacle avoidance and interaction. The authors acknowledge this when noting the benchmark "lacks... the ability to animate other actors" (line 71).
>
> We currently focus on static environments, as most locomotion experiments are typically studied in static scenes [1, 2]. We agree that incorporating dynamic actors would further increase scene complexity and allow for more comprehensive evaluations. This is still compatible with our current system.
>
> In the literature on scene-graph-based dynamic 3DGS, object motion is typically modeled through associated bounding boxes [3, 4]. These methods reconstruct each object in a static canonical frame and simulate dynamics by updating the positions of Gaussians according to the motion of their corresponding bounding boxes. Our system naturally fits similar approach, for example, by assigning Gaussians to dynamic actors (e.g., a soccer ball on the grass) and updating their Gaussian positions over time based on their spatial states in the simulator. This enables Gaussian movement in the rendered views to model dynamic environments.
>
> [1] Xuxin Cheng, et al. Extreme Parkour with Legged Robots. ICRA 2024.
>
> [2] Alan Yu, et al. Learning Visual Parkour from Generated Images. CORL 2024.
>
> [3] Ziyu Chen, et al. OmniRe: Omni Urban Scene Reconstruction. ICLR 2025.
>
> [4] Yunzhi Yan, et al. Street Gaussians for Modeling Dynamic Urban Scenes. ECCV 2024.
>
> **Depth policies are sensitive to depth**
>
> > The benchmark's provided depth data leads to very poor policy performance (Fig. 10-C) , with the authors noting it is "less effective for learning". This is a significant limitation, as depth is a critical modality for locomotion. Other methods, such as in LucidSim, have shown that performance differences between depth and RGB policies are not significant. The extremely poor performance here raises serious questions about the benchmark's depth data quality. The work lacks direct validation (e.g., comparison to ground truth) to prove the data's integrity.
>
> The problem is that depth policies are more sensitive to depth quality and diversity. Lucidsim uses perfect depth from synthetic data while ours are directly from a small set of reconstructed meshes. The domain randomization strategies also differ in our training setting: RGB training involves random rotations, Gaussian blur, and color scaling, while depth training uses random zero-masking and scale variation. RGB inputs naturally offer richer visual features, which support more effective policy learning. These mismatches in augmentation may further contribute to the observed performance gap. We are incorporating more training techniques to improve depth policies and will include them in the final revision.
>
> ##### **We are making our toolchain more scalable!**
>
> > A key weakness, acknowledged by the authors, is that the pipeline requires manual reorientation, rescaling, and task labeling (lines 267-270). This creates a significant bottleneck for scaling the number and diversity of scenes.
>
> We appreciate the reviewer raising this point. In our early version, per-scene annotation took much longer (~15 minutes) due to limited engineering adaptation. Through multiple iterations of improving our labeling and visualization tools with development of a more intuitive GUI, the annotation time is now reduced to 1–2 minutes per scene, which we believe is already highly efficient and makes large-scale data collection feasible.
>
> That said, we strongly agree that a data collection pipeline with no human involvement would further boost efficiency. By introducing foundation models and powerful VLMs, we believe the current toolchain can evolve into a fully automated pipeline, and we are actively working toward this.

---

> > ### Comment · Area_Chair_C7md · 2025-08-04
> >
> > Dear reviewer,
> > I don't see your final justification and final rating yet (as well as any discussion with the authors).
> > Please update the review form and provide feedback to the authors before the deadline.
> > AC

---

> > ### Comment · Reviewer_kab6 · 2025-08-05
> >
> > Thank you for the rebuttal. My concerns have been addressed. I would like to retain the original score.

---

> > > ### Author Response · Authors · 2025-08-08
> > > **Thank you for your feedback!**
> > >
> > > The authors would like to thank `kab6` for their constructive feedback. In the updated draft, we have included additional evaluation that verifies the integrity of the rendering from our 3DGS reconstruction, which indicates improvement over existing SOTA. Some of these results are presented in comments above.
> > >
> > > Thank you for your thoughtful review!

---

### Official Review · Reviewer_Fynv · 2025-07-01

**Rating:** 4
**Confidence:** 4

**Summary:**

This paper introduces Neverwhere, a new benchmark consisting of over 60 high-fidelity environments for evaluating visual locomotion policies in robots. The dataset is built with 3DGS reconstruction of urban indoor and outdoor scenes and further supports closed-loop simulation using MuJoCo. The authors also provide a complete toolchain that allows users to generate new benchmark environments from raw videos with minimal manual effort.

**Additional Feedback:**

In a word, this paper introduces the first parkour benchmark for outdoor embodied scenes with a 3DGS-based generation pipeline. The idea is promising and the system is well-engineered, but some concerns remain (see "Limitations Weaknesses").  My initial rate is borderline accept.

**Dataset Code Accessibility:**

Yes

**Dataset Code Comments:**

The submission provides an anonymized link to the dataset and accompanying benchmark code, which are accessible and downloadable without restrictions.

**Ethical Comments:**

There are no significant ethical concerns associated with this submission. The Neverwhere benchmark suite is a robotics simulation framework developed from 3D reconstructions of publicly accessible university environments.

**Ethical Considerations:**

No, there are no or only very minor ethics concerns

**Final Justification:**

My concerns have been addressed. I would like to keep the original score.

**Limitations Weaknesses:**

1. The claim that 3DGS-based rendering is superior for evaluation is entirely unsubstantiated. There’s no perceptual fidelity study or error metric (PSNR, SSIM, LPIPS, etc.) comparing rendered views to real images.
2. Justification for simulator choice. Given the existence of alternative platforms such as IsaacGym, CoppeliaSim, Pybullet, Gazebo, the authors should clarify why MuJoCo is preferable for their closed-loop evaluation setup. Or any suggestion for integrating the pipeline into other simulators.
3. The methodological innovation is somewhat limited. The proposed pipeline appears to be largely engineering-oriented, consisting primarily of a straightforward application of  3DGS to construct a dataset. There is minimal novelty in algorithm design, scene modeling, or learning paradigms, which raises concerns about the depth of research innovation in the submission.

**Strengths Contributions:**

1. The benchmark covers multiple locomotion challenges (hurdles, gaps, ramps, stairs) with scene diversity across indoor/outdoor domains.
2. The integration of 3DGS to build physically meaningful environments from raw videos is promising.
3. The benchmark offers multiple observation modalities (RGB, depth, semantics, point cloud), enabling embodied evaluation across a range of perception setups.

---

> ### Author Rebuttal · Authors · 2025-07-31
>
> We sincerely thank the reviewer for the time and effort invested in reviewing our work. Your comments are very helpful, our responses regarding the weakness mentioned are detailed below.
>
> **Our contributions extend beyond just scene modeling**
>
> > The methodological innovation is somewhat limited. The proposed pipeline appears to be largely engineering-oriented, consisting primarily of a straightforward application of 3DGS to construct a dataset. There is minimal novelty in algorithm design, scene modeling, or learning paradigms, which raises concerns about the depth of research innovation in the submission.
>
> The primary goal of our work is to provide well-curated benchmark and a practical and scalable toolchain, that bridges the critical sim-to-real gap in visual policy research.
>
> As acknowledged by reviewer `Xeem` of our work's strength, we provide a valuable testbed that captures the complexity of real-world for parkour tasks and enables users to add environments with ease, forming an important complement to current visual policies that has been limited to RGB inputs. A key enabler for real-world applicable RGB policies (e.g. Lucidsim[1]) is the availability of 3DGS-based evaluation environments. We now significantly scale up these closed-loop evaluation environments with *neverwhere*.
>
> Our contributions and innovations are threefold:
>
> 1. **A complete, open-source real-to-sim pipeline** (noted by reviewer `Kab6`): We present an automated toolchain that reconstructs real-world scenes into collision-accurate mesh + 3DGS pairs, without the need for additional sensors or estimators. Our toolchain (scene modeling, labeling, visualization) is not limited to locomotion, it can be adapted to a wider range of robotics tasks with just some small adaptations.
>
> 2. **The largest 3DGS benchmark for locomotion to date**: We release 60+ ready-to-use scenes. These enable researchers to test visual policies across diverse environments and analyze robot failure modes, helping improve real-world deployments.
>
> 3. **A comprehensive study of current boundaries of 3DGS in policy learning:** Although *neverwhere* is not primarily designed for policy training, we include additional experiments (in experiment section in main paper and supplementary) to explore the feasibility and limitations of training directly on our data. This helps clarify the boundaries of our benchmark, shedding light on future research.
>
> We see our work as a foundational step that supports ongoing advances in sim-to-real transfer, data-collection, and closed-loop evaluation.
>
> [1] Alan Yu, et al. Learning Visual Parkour from Generated Images. CORL 2024.
>
> **Additional results on visual fidelity**
>
> > The claim that 3DGS-based rendering is superior for evaluation is entirely unsubstantiated. There’s no perceptual fidelity study or error metric (PSNR, SSIM, LPIPS, etc.) comparing rendered views to real images.
>
> We reconstruct scenes from varied in-the-wild data, where differences in image quality, resolution, and pose accuracy make metrics like PSNR, SSIM, and LPIPS unreliable for assessing visual quality in robot perception (which often uses low-resolution inputs). These metrics also don’t reflect geometric accuracy.
>
> That said, we agree that reporting these metrics can be useful as a reference. Therefore, we conducted additional experiments on 8 scenes (each with 200–400 images at around 1400×1000 resolution), results are provided below:
>
> |                  | PSNR    | SSIM   | LPIPS  |
> | ---------------- | ------- | ------ | ------ |
> | Ours (w/o depth) | 27.5830 | 0.7981 | 0.1482 |
> | Ours (w/ depth)  | 28.0801 | 0.8262 | 0.1556 |
>
> For qualitative assessment, we also provide processed `.splat` files for 3DGS and `.obj` files for each scene in the dataset.
>
> **Our benchmark is simulator-agnostic**
>
> > Justification for simulator choice. Given the existence of alternative platforms such as IsaacGym, CoppeliaSim, Pybullet, Gazebo, the authors should clarify why MuJoCo is preferable for their closed-loop evaluation setup. Or any suggestion for integrating the pipeline into other simulators.
>
> Thank you for the suggestion. We chose MuJoCo primarily for its simulation accuracy and ease of use in our offline closed-loop training setup. However, this choice does not limit the applicability of our pipeline. Our assets and toolchain are inherently simulator-agnostic, built with lightweight wrappers that can interface with different simulator APIs. We appreciate the reviewer’s point and are actively working on extending support and providing integration guidelines for other simulators.

---

> > ### Comment · Area_Chair_C7md · 2025-08-04
> >
> > Dear reviewer,
> > I don't see your final justification and final rating yet (as well as any discussion with the authors).
> > Please update the review form and provide feedback to the authors before the deadline.
> > AC

---

> > ### Comment · Reviewer_Fynv · 2025-08-04
> > **Reply to rebuttal**
> >
> > Thank you for the rebuttal. My concerns have been addressed. I would like to keep the original score.

---

> > > ### Author Response · Authors · 2025-08-08
> > > **Thank you for your feedback!**
> > >
> > > **Additional Results Show Improved Visual Fidelity Metrics on SOTA Baselines**
> > >
> > > The authors appreciate `Fynv`'s constructive feedback. To add to the discussion on perceptual fidelity metrics, we include the results below for your consideration:
> > >
> > > We compare against Nerfstudio’s high-performing models, `splatfacto` and `splatfacto-mcmc`. And tested the pipeline on two tasks: **scene reconstruction** (trained and evaluated on all images) and **novel view synthesis** (trained on 90%, evaluated on 10%).
> > >
> > > The quantitative results show that MVS initialization and depth supervision improve over existing SOTA methods in both PSNR, LPIPS and SSIM. Qualitatively, we also see a significant improvement in challenging lighting conditions involving high contrast daylight and shadowy regions; and reflective surface. These updated evaluations validates the quality of this benchmark suite.
> > >
> > > We hope these results help further clarify the strengths of our overall contribution. In case `Fynv` find these metrics helpful, we would sincerely appreciate an increased score.
> > >
> > > Thank you for your review!
> > >
> > > ----
> > > `splatfacto`
> > >
> > > | Setting | Recon-PSNR | Recon-SSIM | Recon-LPIPS | NVS-PSNR  | NVS-SSIM  | NVS-LPIPS |
> > > | :------ | :--------- | :--------- | :---------- | --------- | --------- | --------- |
> > > | Default | 32.30      | 0.919      | 0.136       | 30.21     | 0.886     | 0.172     |
> > > | Ours    | **32.52**  | **0.925**  | **0.122**   | **30.32** | **0.891** | **0.161** |
> > >
> > > `splatfacto-mcmc`
> > >
> > > | Setting | Recon-PSNR | Recon-SSIM | Recon-LPIPS | NVS-PSNR  | NVS-SSIM  | NVS-LPIPS |
> > > | :------ | :--------- | :--------- | :---------- | --------- | --------- | --------- |
> > > | Default | 33.38      | 0.944      | 0.094       | **31.02** | **0.915** | 0.127     |
> > > | Ours    | **33.45**  | **0.946**  | **0.089**   | 30.92     | **0.915** | **0.124** |
> > >
> > > 1. **Default**: Original Nerfstudio models (`splatfacto`, `splatfacto-mcmc`)
> > > 2. **Ours**: Applying both OpenMVS-based initialization and depth supervision, as described in Section 4 of the submission

---

### Official Review · Reviewer_Xeem · 2025-07-01

**Rating:** 5
**Confidence:** 4

**Summary:**

The paper introduces the Neverwhere visual parkour benchmark, a collection of 60 high-fidelity digital recreated scenes covering diverse urban structures, used to test real-world visuomotor policies in automated, close-loop simulations. The Neverwhere also provides a data collection toolchain that enables building new evaluation environments directly from uncalibrated images or videos. Lastly, the paper also provides visual parkour policy checkpoints trained directly on the Neverwhere 3D Gaussian environments, offering baseline results to support further research and exploration.

**Additional Feedback:**

Please see the limits for the questions.

Here is a typo:
Line 238: The above experiments do not **use** include those cones as observation. We further investigate how different observation types affect policy performance.

**Dataset Code Accessibility:**

Yes

**Dataset Code Comments:**

The authors provide the dataset for 60 scenes included in Neverwhere benchmark. The codebase also include the complete toolchain for creating custom environments from the Polycam Scans. The toolchain relies on open-sourced tools such as COLMAP and OpenMVS.

**Ethical Considerations:**

No, there are no or only very minor ethics concerns

**Final Justification:**

The new evaluation pipeline and the provided toolchain is valuable. However, I do believe more realistic environments beyond those can be handcrafted by 3D meshes should be included in the benchmark to demonstrate the usefulness of the evaluation pipeline. These environments can be brick roughness, gaps between bricks, or irregular rocky paths as pointed by the author. Therefore, I would like to keep the current evaluation score.

**Limitations Weaknesses:**

1. Prior work on visual locomotion has been pretty successful in deploying to the real world with depth images and manually crafted parkour environments based on 3D meshes. See the papers [1, 2]. The scenes created in Neverwhere do not go beyond those that have been explored in the literature, such as hurdles, gaps, ramps. I doubt if these visual parkour tasks actually need to
* Reconstruction scenes from real-world. (Manually crafted 3D meshes with randomness is sufficient to cover real-world complexity.)
* Use photorealistic RBG image. (Depth images might be sufficient for visual parkour tasks.)
Could authors be more specific about the real-world complexities that are not captured by transitional 3D meshes and depth images, but the Neverwhere benchmark?

[1] Agarwal, Ananye, et al. "Legged locomotion in challenging terrains using egocentric vision." Conference on robot learning. PMLR, 2023.

[2] Luo, Shixin, et al. "Pie: Parkour with implicit-explicit learning framework for legged robots." IEEE Robotics and Automation Letters (2024).

2. It is unclear if the physical parameters align with the visual features when creating these scenes. For example, the grass surface is supposed to have higher restitution and friction. The wooden surface might be more slippery.  If not, it is actually counter-intuitive that RGB performs better than depth in Figure 10.

3. The close-loop training setting is also a bit unclear in section 5.1. Extreme parkour was created in IsaacGym with GPU-acceleration, which current MuJoCo does not support. What is the training budge used for single and multi-scene training? Do they match with the existing visual parkour training in the literature?

**Strengths Contributions:**

Previous research relies exclusively on trial-and-error and empirical tricks for successfully deploying visuomotor policies to the real world. There lacks a good testbed in the simulation that captures the complexities of visual parkour tasks in the real world and can be used as a good indicator of the real-world deployment performance. To this end, this paper certainly present an important supplementary to the literature.

Previously in the literature, visuomotor policies have been primarily trained using depth images instead of RGB images rendered from photorealistic scenes. It is also an important addition by comparing Depth visual policies and RGB visual policies.

---

> ### Author Rebuttal · Authors · 2025-07-31
>
> We thank the reviewer for their detailed feedback and positive evaluation of our work. We provide responses to the questions raised in the "Weakness" section below.
>
> **RGB inputs reveal real-world complexities missed by handcrafted depths**
>
> > The scenes in Neverwhere do not go beyond those that have been explored by previous setups, such as hurdles, gaps, ramps. I doubt if these visual parkour tasks actually need to reconstruction scenes from real-world or use photorealistic RBG image.
>
> As the reviewer pointed out in the strengths section, visual-locomotion policies have been limited to RGB inputs. Recent works that use generative models to generate RGB images for visual policy training have shown impressive real-world results (e.g., LucidSim [1]), in part enabled by 3DGS-based simulation for evaluation. *Neverwhere* is our attempt to significantly increase the scale and diversity of such evaluation setups. This scale and diversity require high-quality reconstruction from in-the-wild data, which is often impractical with traditional handcrafted meshes, for example, capturing details like brick roughness, gaps between bricks, or irregular rocky paths. Our toolchain and datasets therefore remain impactful for locomotion research.
>
> **We aim to target a broader range of tasks and domains**
>
> Our toolchain and experience are not limited to the parkour. They can be adapted to other domains with minimal adjustments. We have uncovered many practical and usability challenges (such as collecting high-quality meshes and depth-aligned 3DGS without relying on any additional depth estimators or sensors), which are essential for scalable 3DGS data collection in robotics. We see this as a first step toward broader applications of 3DGS-enabled policies, including future integration with diffusion policies for manipulation.
>
> We agree that including a broader set of domains beyond the task types in our initial submission would strengthen the work. We have already collected new scenes featuring highly irregular terrains (e.g., rocky, mountainous roads), and will include them in our final revisions.
>
> [1] Alan Yu, et al. Learning Visual Parkour from Generated Images. CORL 2024.
>
> **Depth policies are sensitive to depth**
>
> > It is unclear if the physical parameters align with the visual features when creating these scenes. If not, it is actually counter-intuitive that RGB performs better than depth in Figure 10.
>
> Our benchmark focuses on visual gap, physical parameters remained a uniform value for each scene. Regarding the performance difference between RGB- and depth-based policies: both use identical physical parameters, so the gap is not from physics. The problem is that depth policies are more sensitive to depth quality and diversity. Our policies are trained on depth from a small set of reconstructed meshes. In contrast, RGB inputs naturally offer richer visual feature and works better on low-scale data. And the domain randomization strategies also differ: RGB training involves random rotations, Gaussian blur, and color scaling, while depth training uses random zero-masking and scale variation. These mismatches in augmentation may further contribute to the observed performance gap. We are incorporating more training techniques to improve depth policies and will include them in the final revision.
>
> **More details about training, implementation, etc**
>
> > Extreme parkour was created in IsaacGym with GPU-acceleration, which MuJoCo does not support. What is the training budge used for training? Do they match with the existing visual parkour training in the literature?
>
> Thanks for the feedback. We will include more experimental details in the revised version.
>
> We chose MuJoCo primarily for its simulation accuracy and ease of use in our offline, closed-loop training setup. This choice is consistent with prior methods that train RGB visual policies in locomotion domains [1].
>
> Each policy takes approximately 4 days to train on a single A100 GPU, covering 4,000 student rollouts and 8,000 epochs of policy training. To accelerate the process, we use the Ray [2] library to parallelize multiple rollout environments on a single GPU.
>
> Although our current training uses MuJoCo, our benchmark and toolchain are simulator-agnostic and can be adapted to other simulators by implementing custom rendering wrappers. We hope this flexibility broadens the applicability of our work across domains.
>
> [1] Alan Yu, et al. Learning Visual Parkour from Generated Images. CORL 2024.
>
> [2] Ray: https://github.com/ray-project/ray

---

> > ### Comment · Reviewer_Xeem · 2025-08-04
> > **Thank you for the clarification**
> >
> > Thank you for the clarification! I would like to keep the original score.

---

> > > ### Comment · Area_Chair_C7md · 2025-08-04
> > >
> > > Ok, but you have to update the form as per instructions.
> > > AC

---

> > > ### Author Response · Authors · 2025-08-08
> > > **Thank you for your feedback!**
> > >
> > > The authors appreciate reviewer `Xeem`'s constructive feedback, and would like to thank them for their time and effort.

---

> ### Comment · Area_Chair_C7md · 2025-08-04
>
> Dear reviewer,
> I don't see your final justification and final rating yet. Please update the review form and provide feedback to the authors by the deadline.
> AC

---

### Official Review · Reviewer_iWLo · 2025-07-02

**Rating:** 4
**Confidence:** 4

**Summary:**

This paper considers a problem of training & testing of visual locomotion policies for the legged robots. The goal of the authors is to narrow the gap between robot training on simulated data & evaluation of their performance in complex real-world environments. As a solution the authors propose to use 3DGS-based reconstructions of real environments in a closed-loop simulator. The authors have developed a framework to create new environments from RGB video, based on COLMAP library for camera pose estimation and 3DGS for 3D scene reconstruction. In order to improve geometry of the resulting environments for better physics simulation and better depth map rendering for depth-based policy training the authors propose to use OpenMVS library for multi-view geometry reconstruction. Using the developed framework a dataset of 60 indoor and outdoor scenes is created. To validate the proposed benchmark and simulation framework 2 types of experiments have been made - single-scene training and multi-scene training.

**Additional Feedback:**

In abstract it is claimed that in this work the developed evaluation environment are "hyper-photo-realistic", which is not supported by the figures with examples of rendered images from the simulator.

**Dataset Code Accessibility:**

Yes

**Ethical Comments:**

The dataset captures some indoor & outdoor environment, and capturing of such data has no ethical issues.

**Ethical Considerations:**

No, there are no or only very minor ethics concerns

**Final Justification:**

After considering the detailed author's answers to my questions and other reviews, I still think that the proposed 3DGS pipeline is outdate in comparison with existing 3DGS methods. However, it seems to be the best currently existing to the specified task. And considering that it the first approach to capturing real-world data to the specific locomotion task, overall it raises significance and impact of work. So I'm changing my rating from 2 to 4.

**Limitations Weaknesses:**

1) Limited novelty.
* 3DGS representation & visualization hasn't been applied specifically to locomotion policies training & bechmarking, but such methods are extensively researched and used for creation of driving closed-loop simulators. 3DGS-based driving simulators are already quite advances, with high visual quality and diverse features, such as handling of dynamic objects, scene editing, etc. (e.g. https://github.com/ziyc/drivestudio) The difference between robotic simulators and driving simulators should be specifically addressed. Also for robotics tasks, there are already works based on 3DGS, e.g. for manipulation (https://arxiv.org/abs/2409.20291) or navigation (https://arxiv.org/abs/2403.02751)
* Authors highlight that they propose to use OpenMVS to improve geometry representation compared to available 3D reconstruction applications. However, the usage of depth maps with 3DGS has already been extensively researched in order to improve geometry and to guide 3DGS-based reconstruction for better speed and visual quality (e.g. https://xuqianren.github.io/ags_mesh_website/, https://maturk.github.io/dn-splatter/)

2) Limited impact and significance:
* The proposed framework for 3DGS-based scene reconstruction for locomotion closed-loop simulator is outdated in terms of used 3DGS methods, which results in poor visual quality, speficially in the far regions. But it could definetely affect the "sim-to-real" gap and lead to poor usefullness of the proposed dataset and benchmark.
* Currently, the proposed dataset provides little for achieving the stated goal of narrowing the train&test gap. As authors mention  - "This suggests that the trained visual policies exhibit limited generalization on our benchmark, particularly for more challenging tasks."
* Overal, the authors haven't provided a clear answer and experimentat validation for usefulness of the proposed dataset & closed-loop simulator as benchmark. One could think that closed-loop simulator should be a good proxy for testing the learned policies in real-world environments.  However, the only experiment of this kind, provided in "5.4 Evaluating Visual Parkour Policies" is totally unconvincing. On the provided dataset Lucidsim shows 59.67% on hurdle and 55.82% on stares. And authors say " This aligns roughly with Lucidsim’s reported results of 73.3% for hurdles and 100% for stairs". I disagree. This results are totally unaligned.

**Strengths Contributions:**

1) Novelty. To the best of my knowledge, this is the first work to apply 3DGS-based reconstruction for simulation for legged robot locomotion.
2) Potential impact. 3DGS-based reconstructions and visualization, which is usually have highter visual realism compared to traditional CG-based rendering engines, can potentially limit the gap between sim2real transfering of learning robot policies.
3) Extensibility. Authors are providing a framework for processing new videos and creation of new environments, which allow users to easily add new scenes to the dataset.
4) Clear writing and paper organization. Overall, the paper is well organized. Key ideas are easy to understand and they are clearly presented.

---

> ### Author Rebuttal · Authors · 2025-07-31
>
> The authors are grateful for the reviewer's constructive feedback. Our responses are detailed below.
>
> **On Novelty: From Autonomous Driving to Contact-Rich Robots Domains**
>
> As robot foundation model efforts begin to scale in a way that is parallel to those of large language models (LLMs), scalable and fully automated closed-loop evaluation has become an essential component of the data flywheel. So far, works in robotics that uses 3DGS as a scene representation primarily focus on training. Our goal for this paper is to show that 1) 3DGS is insufficient diversity-wise to produce generalizable policies, and 2) it is a perfect stack for closed-loop evaluation. And in the long run, we consider this to be the first 3DGS benchmark suite for contact-rich problems, and are planning on building manipulation benchmarks upon this foundation.
>
> The overall agenda is to push robotics forward by developing better evaluation methods.
>
> ***getting accurate geometry is harder than we thought***
>
> The authors are intimately familiar with drivestudio. Whereas driving simulators focus on getting accurate appearance and models of moving viechles and pedestrians, most robotics tasks that we consider difficult involve complex physical contact that depends on the geometry. Our experience scaling up 3DGS scanning for locomotion showed that methods that otherwise gets good performance on benchmark datasets do not produce good geometry even with reasonably carefully taken videos.
>
> **SOTA 3DGS methods, in practice, reconstruct geometries poorly**.
>
> In particular, under direct sunlight where there is a high contract, shadow area struggles with getting accurate geometry. Our openMVS + 3DGS pipeline addresses this problem. We will include targeted renders and examples in the cameraready to better present this. Second, cellphone videos taken indoors usually have high ISO settings, where the denoising algorithms erase fine texture details. We found that carpeted floors are very difficult to reconstruct. We also had issues with reflective marble surface, etc. Our MVS -> 3DGS pipeline great improves the reconstruction in these situations.
>
> **Driving sims do not use MuJoCo; Modeling larger 3DGS scenes while maintaining geometric details is challenging**
>
> Getting accurate geometric details in larger scenes (ones we present) turn out to be quite challenging. So the majority of our technical innovations focuses these practical problems. These contributions are not very glamorous, but we believe they are quite befitting for the dataset and benchmark track.
>
> **A Clarification: Why Not Use Depth/Normal Priors?**
>
> > Authors highlight that they propose to use OpenMVS to improve geometry representation compared to available 3D reconstruction applications. However, the usage of depth maps with 3DGS has already been extensively researched in order to improve geometry and to guide 3DGS-based reconstruction for better speed and visual quality (e.g. https://xuqianren.github.io/ags_mesh_website/, https://maturk.github.io/dn-splatter/)
>
> We spent significantly time before and during this project with prior-assisted 3DGS methods. OpenMVS gave significantly better mesh. Our general oppinion is that mesh is a much more suitable representation format for geometry than 3DGS in part because radiance gaussian mixtures are way too expressive both spatially (it overfits to minor camera pose errors, and currently there is no good solution to this problem); and geometrically. in our experience OpenMVS can produce smooth, and GT-like surface reconstructions, whereas if you look at results from DN-Splatter, the carpets if all of those indoor scenes appear rough. Althought we can not supply additional figures in this rebuttal, we have already added comparison against prior-assisted 3DGS setups in the updated draft.
>
> **Going from images to 3DGS to geometry is backwards -- just get the geometry first, and use it to help 3DGS training**.
>
> The place where priors excel are plain, texture-less surfaces. When the surface has texture, depth and normal priors often hallucinate surface features following the luminance changes in the texture. Both AGS-Mesh [1] and DN-Splatter [2] suffers from this problem.
>
> [1] Ren, et al. AGS-Mesh: Adaptive Gaussian Splatting and Meshing with Geometric Priors for Indoor Room Reconstruction Using Smartphones. 3DV 2025.
>
> [2] Turkulainen, et al. DN-Splatter: Depth and Normal Priors for Gaussian Splatting and Meshing. WACV 2025.
>
> ### So is our 3DGS method outdated? -- No.
>
> > The proposed framework for 3DGS-based scene reconstruction for locomotion closed-loop simulator is outdated in terms of used 3DGS methods, which results in poor visual quality, speficially in the far regions. But it could definetely affect the "sim-to-real" gap and lead to poor usefullness of the proposed dataset and benchmark.
>
> Since robotic tasks require robust mesh reconstruction from complex real-world scenes, we use OpenMVS and build on its output to improve 3DGS quality. This results in a practical, up-to-date pipeline suited for robotics. Meanwhile, our 3DGS module is modular, newer techniques can be easily integrated. We are not proposing a novel graphics method, but rather a realistic, extensible evaluation testbed for visual parkour policies (as also noted by reviewer `Xeem`). This toolchain along with the data we collected, capture real-world complexity and makes it easy to add new environments, forming an important complement to current visual policies that has long been limited to RGB inputs.
>
> **Question about closed-loop evaluation results**
>
> Since the tested scenes differ between LucidSim and our experiments, their results are not directly comparable (that's why we say *roughly aligns* in in the draft). The reported scores of 59.67% and 55.82% represent averages over dozens of simulated scenes, including both very challenging and simple cases. In contrast, real-robot experiments typically involve simpler setup, for example, LucidSim’s hurdle experiment includes only one hurdle, while most of our simulated scenes use three hurdles. A more meaningful comparison lies in analyzing how the robot behaves in simulation vs. the real world, especially in failure cases where similar patterns emerge. We have done this comparison and will include more figure and video illustrations in the updated draft.

---

> > ### Comment · Reviewer_iWLo · 2025-08-04
> >
> > I thank authors for providing a detailed answer to my key concerns. While I still think that in terms of technical novelty the proposed 3DGS pipeline is outdated, it it totally possible that for the considered task it is currently better suited then other existing approaches. Taking into account the detailed feedback and notes from my fellow reviewers I'm considering raising my score.

---

> > ### Author Response · Authors · 2025-08-08
> > **Thank you for your feedback!**
> >
> > **MVS Init + Depth Supervision Consistently Improves Nerfstudio Baselines**
> >
> > The authors would like to thank `iWLo` for considering raising the score. Since submission, we connected our MVS-based initialization scheme and the resulting depth-supervision (in Section 4 of the submission) to the newest version of nerfstudio. It appears our method -- getting the geometry **first** from MVS, and then use it to guide 3DGS **after**, consistently improved the performance of two of the top-performing GS methods -- `splatfacto-mcmc` and `splatfacto`.
> >
> > On `splatfacto`, PSNR went from 32.30 -> 32.52; on `spatfacto-mcmc` it went from 33.38 -> 33.45. Besides the improved metrics, we also notice that our method consistently improved the perceptual quality in low-light areas in high-contrast lighting conditions. We have updated the draft on our end, and plan to include these exmaples in the cameraready upon acceptance.
> >
> > **Thank you!**
> >
> > Please let us know if this addresses earlier concerns, and if an improved score is appropriate. The authors appreciate iWLo's constructive feedback deeply.
> >
> > -------
> >
> > **experiment details**: we connected our schemes with `splatfacto` and `splatfacto-mcmc`. We tested the following four setups:
> >
> > 1. **Original model** (splatfacto, splatfacto-mcmc)
> > 2. **OpenMVS depth**: using patch-matched depth from OpenMVS for depth supervision
> > 3. **OpenMVS initialization**: replacing SFM points with points sampled from colored meshes generated by OpenMVS
> > 4. **Combined**: applying both initialization and depth supervision
> >
> > We tested on two tasks: Scene Reconstruction (trained and evaluated on all images) and Novel View Synthesis (trained on 90%, evaluated on 10%). Average results across a batch of 7 scenes are reported below:
> >
> > `splatfacto`
> >
> > | Setting             | Recon-PSNR | Recon-SSIM | Recon-LPIPS | NVS-PSNR  | NVS-SSIM  | NVS-LPIPS |
> > | :------------------ | :--------- | :--------- | :---------- | --------- | --------- | --------- |
> > | original            | 32.30      | 0.919      | 0.136       | 30.21     | 0.886     | 0.172     |
> > | +depth              | 32.39      | 0.922      | 0.128       | 30.21     | 0.888     | 0.165     |
> > | **ours 👇**          |            |            |             |           |           |           |
> > | +openmvs_init       | 32.40      | 0.921      | 0.132       | **30.34** | 0.889     | 0.169     |
> > | +depth+openmvs_init | **32.52**  | **0.925**  | **0.122**   | 30.32     | **0.891** | **0.161** |
> >
> > `splatfacto-mcmc`
> >
> > | Setting             | Recon-PSNR | Recon-SSIM | Recon-LPIPS | NVS-PSNR  | NVS-SSIM | NVS-LPIPS |
> > | :------------------ | :--------- | :--------- | :---------- | --------- | -------- | --------- |
> > | original            | 33.38      | 0.944      | 0.094       | 31.02     | 0.915    | 0.127     |
> > | +depth              | 33.32      | 0.944      | 0.094       | 30.90     | 0.915    | 0.126     |
> > | **ours 👇**          |            |            |             |           |          |           |
> > | +openmvs_init       | **33.47**  | 0.945      | **0.089**   | **31.07** | 0.915    | **0.124** |
> > | +depth+openmvs_init | 33.45      | **0.946**  | **0.089**   | 30.92     | 0.915    | **0.124** |
> >
> > In more challenging scenes—such as those with shadows or reflective surfaces—3DGS methods without explicit improvements to scene geometry often produce incorrect surfaces. Our proposed methods, both in the original version and in the current Nerfstudio integration, can significantly reduce these artifacts. This is essential for scaling up 3dgs evaluation.

---

> ### Comment · Area_Chair_C7md · 2025-08-04
>
> Dear reviewer,
> your original recommendation was "reject". The authors provided several feedbacks. Please engage in discussion with them and provide your final recommendation by the deadline.
> AC

---

### Note · Authors · 2025-08-15

The authors sincerely thank AC and all reviewers for their responsiveness and kind reviews.

Notably, they acknowledge that this work:

> Provide an important framework and testbed for visual policy learning and evaluation.  (`Xeem`, `kab6`)

> Developed an extensible, scalable toolchain for in-the-wild data. Sim supports multi-modal observations. (`iWLo`, `kab6`, `Fynv`)

> The resulting 3DGS environments meaningfully help bridge the sim-to-real gap in locomotion. (`iWLo`, `Fynv`)


Our rebuttal further addressed the following reviewer concerns:

**Novel Method Tailored Towards Robotic Domain**

3DGS is powerful for modeling scene appearance, but it alone is insufficient to support physical robot simulation. We discovered a few scaling issues in capturing real-world scenes, and demonstrated significantly better collision geometry reconstruction compare to recent works. Without using lidar/depth priors.

**SOTA in Perceptual Metrics**

Taking reviewer `iWLo` and `Fynv`'s feedback, we upgraded the toolchain to integrate with the newest version of Nerfstudio. Our MVS initialization brought a new state of the art PSNR and perceptual fidelity. (table attached below)

**Results Demonstrate the Value of Neverwhere in Robot Evaluation**

Our position is that 3DGS environments, when done right, offer significant advantage in robot evaluation. We emphasize that the goal is NOT to use neverwhere for training.

Our end-to-end goal is to scale up realistic robot evaluation environments. We address key challenges in large-scale 3D modeling for robotics, ensuring that 3DGS environments directly benefit visual policies.

----
`splatfacto`

|         | Recon-PSNR | Recon-SSIM | Recon-LPIPS | NVS-PSNR  | NVS-SSIM  | NVS-LPIPS |
| :------ | :--------- | :--------- | :---------- | --------- | --------- | --------- |
| Default | 32.30      | 0.919      | 0.136       | 30.21     | 0.886     | 0.172     |
| Ours    | **32.52**  | **0.925**  | **0.122**   | **30.32** | **0.891** | **0.161** |

`splatfacto-mcmc`

|         | Recon-PSNR | Recon-SSIM | Recon-LPIPS | NVS-PSNR  | NVS-SSIM  | NVS-LPIPS |
| :------ | :--------- | :--------- | :---------- | --------- | --------- | --------- |
| Default | 33.38      | 0.944      | 0.094       | **31.02** | **0.915** | 0.127     |
| Ours    | **33.45**  | **0.946**  | **0.089**   | 30.92     | **0.915** | **0.124** |

**Default**: Original Nerfstudio models.

**Ours**: Applying both OpenMVS-based initialization and depth supervision.

---

### Decision · Program_Chairs · 2025-09-18

**Decision:**

Reject

**Comment:**

This paper introduces a framework and a benchmark for testing visual locomotion policies. The goal is to to encourage large-scale and reproducible robot evaluation by making it easier to create and integrate Gaussian splats-based reconstructions. This is a relevant and timely task and it has been appreciated by the reviewers. However, the paper originally got mixed reviews with some serious concerns about the proposed 3DGS pipeline and missing comparisons (especially form Reviewer iWLo).

The authors provided several feedbacks, clarification and new results. The paper has been discussed extensively and, after the rebuttal phase, there is a general consensus towards acceptance. The major criticisms have been adequately addressed, and the most critical reviewers raised their original score. The AC believes the reasons to accept the paper outweigh reasons to reject, and strongly encourages the authors to include the material presented in the rebuttal phase in their camera ready.

===== FINAL UPDATE FROM DB Track PCs ====

The final decision for this paper has been taken by the program chairs after consultation with the SACs. All Senior Area Chairs have ranked papers according to the feedback from the AC during the review process. We decided to leave the original meta-review to reflect the opinion of the AC in light of the initial discussions with reviewers and SAC.